# TRAIN ON VALIDATION (ToV):
# FAST DATA SELECTION WITH APPLICATIONS TO FINE-TUNING

**Ayush Jain**[*]
Google Research
ayush.jain.2508@gmail.com

**Eren Sasoglu**[*]
Encord
eren.sasoglu@gmail.com

**Andrea Montanari**[†]
Granica Computing Inc.– (granica.ai)
andrea.montanari@granica.ai

## ABSTRACT

State-of-the-art machine learning often follows a two-stage process: $(i)$ pre-training on large, general-purpose datasets; $(ii)$ fine-tuning on task-specific data. In fine-tuning, selecting training examples that closely reflect the target distribution is crucial. However, it is often the case that only a few samples are available from the target distribution. Existing data selection methods treat these target samples as a validation set and estimate the effect of adding or removing a single sample from the training pool by performing inference on the validation set.

We propose a simpler and faster alternative that inverts the usual role of train and validation: we perform inference on the training pool before and after fine-tuning on the validation set. We then select samples whose predictions change the most. Our key insight is that the training samples most affected by fine-tuning on a small validation set tend to be the most beneficial for reducing test loss on the target distribution. Experiments on instruction tuning and named entity recognition tasks show that, in most cases, our method achieves lower test log-loss than state-of-the-art approaches. We support our findings with theoretical analysis.

## 1 INTRODUCTION

While large language models (LLMs) are pretrained on internet-scale datasets, their downstream performance can be heavily dependent on the instruction-tuning stage in which they are fine-tuned on instruction/output pairs (Ouyang et al., 2022; Zhou et al., 2024; Longpre et al., 2023; Chung et al., 2024). These datasets are significantly smaller and are often gathered by using multiple heterogeneous sources. Instruction tuning becomes even more difficult when targeting a specialized use case (Wang et al., 2023). More generally, scarcity of domain-specific data is a ubiquitous challenge when fine-tuning foundation models.

This paper presents an easy-to-implement and low-complexity method for selecting a training dataset of prescribed size from heterogeneous sources to maximize the test time performance on the target distribution. Our method is motivated by the theory of influence functions (van der Vaart, 2000) yet avoids the computational burden of computing influence functions. Unlike influence-function–based data selection methods, ToV avoids per-example gradients, Hessian–vector products, and backpropagation through training dynamics. It instead estimates example utility using only forward loss evaluations before and after a small gradient step on the validation data. We validate this approach on two token-based learning tasks, instruction tuning and named entity recognition (NER), and show that in most cases it outperforms state-of-the-art data selection baselines. To illustrate its broad applicability, we show that it yields interesting results even for a simple logistic regression example (see Appendix D).

---

[*]Work done while at Granica Computing Inc.
[†]Department of Statistics and Department of Mathematics, Stanford University

To formalize the problem, assume access to two datasets: a small dataset from the target distribution $\mathbb{P}$ on $\mathcal{Z}$ and a larger one from possibly heterogeneous data sources. We refer to the dataset from the target as the 'validation set' $\boldsymbol{Z}^{\text{val}} := (\boldsymbol{z}_1^{\text{val}}, \ldots, \boldsymbol{z}_{m_{\text{val}}}^{\text{val}})$ where $\boldsymbol{z}_i^{\text{val}}$ are i.i.d. samples from the target distribution $\mathbb{P}$, and to the larger heterogeneous dataset as 'training pool' $\boldsymbol{X} = (\boldsymbol{x}_1, \ldots, \boldsymbol{x}_N)$, where $\boldsymbol{x}_i \in \mathcal{Z}$. In general the distribution of the training pool differs from $\mathbb{P}$. Our goal is to minimize the test error on the target distribution with respect to the model parameters $\boldsymbol{\theta} \in \mathbb{R}^p$:

$$R(\boldsymbol{\theta}) := \mathbb{E}[\ell(\boldsymbol{\theta}, \boldsymbol{z})],\tag{1}$$

where $\ell : \mathbb{R}^p \times \mathcal{Z} \to \mathbb{R}$ is a loss function. A separate target-distribution test set $\boldsymbol{Z}^{\text{tst}}$ (separate from $\boldsymbol{Z}^{\text{val}}$) is used to estimate $R(\boldsymbol{\theta})$ after fine-tuning.

We aim to achieve this by training (or fine-tuning) the model on a subset $S \subseteq [N]$ of the training pool, e.g. running stochastic gradient descent (SGD) with respect to the empirical risk:

$$\widehat{R}_S(\boldsymbol{\theta}) := \frac{1}{|S|} \sum_{i \in S} \ell(\boldsymbol{\theta}, \boldsymbol{x}_i).\tag{2}$$

Let $\hat{\boldsymbol{\theta}}_S$ be the outcome of running SGD (or any specific training algorithm) on $\widehat{R}_S(\boldsymbol{\theta})$. We want to select the subset $S$ (given a constraint on its size $|S|$) so that $\hat{\boldsymbol{\theta}}_S$ achieves a small test loss on the target distribution, i.e. as to minimize $R(\hat{\boldsymbol{\theta}}_S)$.

## 1.1 TRAIN ON VALIDATION: MOTIVATION AND ALGORITHM

To select the most helpful examples at model $\boldsymbol{\theta}$, we might score training examples by the decrease in validation loss induced by a single gradient step with respect to that example, then select those with the highest scores. Computing these scores directly requires $N+1$ full evaluations over the validation set. We derive an efficient approximation to these scores.

Consider a single gradient step with respect to a training example $\boldsymbol{x}$:

$$\boldsymbol{\theta}_{\boldsymbol{x}} = \boldsymbol{\theta} - \eta \nabla \ell(\boldsymbol{\theta}, \boldsymbol{x}).\tag{3}$$

The corresponding change in loss for a validation example $\boldsymbol{z}$, $\ell(\boldsymbol{\theta}, \boldsymbol{z}) - \ell(\boldsymbol{\theta}_{\boldsymbol{x}}, \boldsymbol{z})$, can be approximated by a first-order Taylor expansion:

$$\ell(\boldsymbol{\theta}, \boldsymbol{z}) - \ell(\boldsymbol{\theta}_{\boldsymbol{x}}, \boldsymbol{z}) \approx -\langle \nabla \ell(\boldsymbol{\theta}, \boldsymbol{z}), \boldsymbol{\theta}_{\boldsymbol{x}} - \boldsymbol{\theta} \rangle = \eta \langle \nabla \ell(\boldsymbol{\theta}, \boldsymbol{z}), \nabla \ell(\boldsymbol{\theta}, \boldsymbol{x}) \rangle,\tag{4}$$

where the last step follows from Eq. (3). Pruthi et al. (2020) approximate the scores by computing gradients for each training and validation example and taking their dot products; Xia et al. (2024) extend this to token-based learning. Our method diverges from these approaches: it requires no per-example gradients.

Note the right-hand side is symmetric in $\boldsymbol{x}$ and $\boldsymbol{z}$. In other words, the decrease in loss on $\boldsymbol{z}$ from a step on $\boldsymbol{x}$ is mirrored by the decrease in loss on $\boldsymbol{x}$ from a step on $\boldsymbol{z}$. Our method exploits this train–validation symmetry. The change in overall validation loss for a single gradient step with respect to $\boldsymbol{x}$ is:

$$\frac{1}{m_{\text{val}}} \sum_{i=1}^{m_{\text{val}}} \left( \ell(\boldsymbol{\theta}, \boldsymbol{z}_i) - \ell(\boldsymbol{\theta}_{\boldsymbol{x}}, \boldsymbol{z}_i) \right) \approx \frac{1}{m_{\text{val}}} \sum_{i=1}^{m_{\text{val}}} \eta \langle \nabla \ell(\boldsymbol{\theta}, \boldsymbol{z}_i), \nabla \ell(\boldsymbol{\theta}, \boldsymbol{x}) \rangle.\tag{5}$$

On the other hand, performing a batch gradient step at $\boldsymbol{\theta}$ with respect to the validation set gives $\boldsymbol{\theta}_{\boldsymbol{Z}^{\text{val}}} = \boldsymbol{\theta} - \eta \frac{1}{m_{\text{val}}} \sum_{i=1}^{m_{\text{val}}} \nabla \ell(\boldsymbol{\theta}, \boldsymbol{z}_i)$. Combining this with Eq. (5), we get

$$\frac{1}{m_{\text{val}}} \sum_{i=1}^{m_{\text{val}}} \left( \ell(\boldsymbol{\theta}, \boldsymbol{z}_i) - \ell(\boldsymbol{\theta}_{\boldsymbol{x}}, \boldsymbol{z}_i) \right) \approx \langle \boldsymbol{\theta} - \boldsymbol{\theta}_{\boldsymbol{Z}^{\text{val}}}, \nabla \ell(\boldsymbol{\theta}, \boldsymbol{x}) \rangle \approx \ell(\boldsymbol{\theta}, \boldsymbol{x}) - \ell(\boldsymbol{\theta}_{\boldsymbol{Z}^{\text{val}}}, \boldsymbol{x}).\tag{6}$$

In other words, the change in average validation loss from training on $\boldsymbol{x}$ can be approximated by the change in loss on $\boldsymbol{x}$ after training on the validation set $\boldsymbol{Z}^{\text{val}}$.

The approximation in Eq. (6) follows directly from a first-order Taylor expansion and therefore relies only on the update induced by the validation step being small and the loss landscape being locally

---

**Algorithm 1** ToV Scoring Algorithm: *Interleaved ToV.*

---

1: **Input:** Pretrained model $\boldsymbol{\theta}_0$, validation set $\boldsymbol{Z}^{\text{val}}$, training pool $\boldsymbol{X} = (\boldsymbol{x}_i : i \in [N])$, epochs $L$,
2:        selected data count $n$, learning-rate schedule $\{\eta_k\}_{k=1}^L$, base model count $m, \varepsilon \in [0, 1)$
3: **Output:** Set of examples $S \subset [N]$ of size $n$
4: Sample base subset $U \subseteq [N]$ of size $m$ randomly; define $\boldsymbol{X}_U = (\boldsymbol{x}_i : i \in U)$
5: Initialize model: $\hat{\boldsymbol{\theta}}_0^{\text{bas}} \leftarrow \boldsymbol{\theta}_0$; set scores $\phi_i \leftarrow 0$ for all $i \in [N] \setminus U$
6: **for** $k = 1$ to $L$ **do**
7:      Train $\hat{\boldsymbol{\theta}}_{k-1}^{\text{bas}}$ on $\boldsymbol{X}_U$ for one epoch with learning rate $\eta_k$ to obtain $\hat{\boldsymbol{\theta}}_k^{\text{bas}}$
8:      Train $\hat{\boldsymbol{\theta}}_k^{\text{bas}}$ for one epoch on $\boldsymbol{Z}^{\text{val}}$ with a learning rate $\varepsilon\eta_k$ to obtain $\hat{\boldsymbol{\theta}}_k^{\text{val}}$
9:      **for** each $i \in [N] \setminus U$ **do**
10:         $\phi_i^{(k)} \leftarrow F(\ell(\hat{\boldsymbol{\theta}}_k^{\text{val}}; \boldsymbol{x}_i) - \ell(\hat{\boldsymbol{\theta}}_k^{\text{bas}}; \boldsymbol{x}_i))$ (see Section 2.1 for the definition of $F$)
11:         $\phi_i \leftarrow \phi_i + \phi_i^{(k)}/L$
12:      **end for**
13: **end for**
14: Return set $S \subseteq [N] \setminus U$ of size $n$ on the basis of scores $\phi_i$ (see text)

---

smooth. If the update is too large or the landscape is highly non-linear, the approximation may degrade. In practice, we enforce this regime by using a validation learning rate that is smaller than the base learning rate. No other assumptions, such as independence between training and validation samples, are required.

Our main objective is to evaluate the left-hand side of Eq. (6) for all $\boldsymbol{x}$ in the training set. The right-hand side provides a far more efficient route: $(i)$ Compute the loss $\ell(\boldsymbol{\theta}, \boldsymbol{x})$ for all training examples; $(ii)$ fine-tune $\boldsymbol{\theta}$ on the validation set to obtain $\boldsymbol{\theta}_{\boldsymbol{Z}^{\text{val}}}$; $(iii)$ re-evaluate the new loss $\ell(\boldsymbol{\theta}_{\boldsymbol{Z}^{\text{val}}}, \boldsymbol{x})$ at each training sample $\boldsymbol{x}$, and approximate the effect of training on $\boldsymbol{x}$ by computing the difference with the loss at point $(i)$.

This requires one epoch of training on the validation set and two evaluations over the training pool, as opposed to $N$ evaluations of the validation loss as suggested by a direct evaluation of the left-hand side of Eq. (6), and it does not require access to per-example gradients.

In the next sections, we use this idea to obtain a selection algorithm that alternates training on a subset of the training set and on the validation set. A specific implementation, which we refer to as *Interleaved ToV* (Method A), is given in Algorithm 1; a slightly different implementation, *Parallel ToV* (Method B), is given in Algorithm 2. In Method A, we start with a small random subset $U \subset [N]$ of the training pool. We train on $U$ for $L$ epochs, resulting in models $\hat{\boldsymbol{\theta}}_1^{\text{bas}}, \ldots \hat{\boldsymbol{\theta}}_L^{\text{bas}}$. For each epoch $k \in [L]$ we fine-tune $\hat{\boldsymbol{\theta}}_k^{\text{bas}}$ for one epoch on the validation set, resulting in models $\hat{\boldsymbol{\theta}}_k^{\text{val}}$. For each epoch, every remaining training example $\boldsymbol{x}_i$ with $i \in [N] \setminus U$ is scored by the change in its loss between $\hat{\boldsymbol{\theta}}_k^{\text{bas}}$ and $\hat{\boldsymbol{\theta}}_k^{\text{val}}$, and scores are averaged across epochs.

The names reflect the mechanics: *Interleaved ToV* interleaves base-set training with a single validation update at each checkpoint, whereas *Parallel ToV* runs two training trajectories in parallel, only one of which is updated on the validation set (see Section 3).

After computing scores $\phi_i$ as in Algorithm 1, we select $S$ using one of two strategies: $(i)$ choose the $n$ examples with the largest $\phi_i$; $(ii)$ choose half from the highest-scoring examples and the other half uniformly at random from $U$ to increase diversity.

Intuitively, large $\phi_i$ means that a small amount of training on the target distribution produces a large change in the model output at $\boldsymbol{x}_i$. Our working assumption, motivated by the heuristics above and formalized in Section 3, is that the converse also holds: training on $\boldsymbol{x}_i$ will produce a large change in model output on the target distribution. Hence the scores $\phi_i$ can be used to select 'important' samples for the target.

An adaptation for token-based learning is described in Section 2, along with empirical results. Section 3 provides a mathematical justification that formalizes the argument above.

## 1.2 RELATED WORK

Our work relates to data selection and data attribution. The impact of a single example on the validation error can be approximated by a first-order Taylor expansion. This idea results in data selection methods based on influence functions (Wang et al., 2018; 2020; Ai et al., 2021; Kolossov et al., 2024). Classical influence functions estimate the effect of a single example on the empirical risk minimizer. Most closely related to our work are Pruthi et al. (2020); Bae et al. (2024); Xia et al. (2024), which instead estimate the influence of an example on the training dynamics. In particular, Bae et al. (2024) shows how to approximately propagate gradient changes at $k$-th epoch through all subsequent epochs. In contrast, Pruthi et al. (2020); Xia et al. (2024) make a crude approximation for this propagation. Limitations of influence-based methods are discussed in Schioppa et al. (2023).

Related ideas also were investigated in Liu et al. (2018), which however computes hypergradients for architecture search, rather than deriving per-example data-selection scores as we do. Concurrent work (Savani et al., 2025) also exploits a related directional-derivative symmetry, but in a different setting (token-level antidistillation) rather than per-example data selection.

The recent work of Xia et al. (2024) proposes LESS, a data selection method for instruction tuning that adapts influence ideas to Adam and long sequences. In particular, these authors emphasize the challenge of computing and storing gradients to compute influences. They address this problem via random projections and low-rank approximation. Engstrom et al. (2024) apply the datamodel framework (Ilyas et al., 2022; Park et al., 2023) to select pretraining data. Separately, a *replay* algorithm that stores only a logarithmic number of checkpoints is proposed in Engstrom et al. (2025). Methods that align training data distributions to a small target set include TSDS (Liu et al., 2024) and DSIR (Xie et al., 2023); domain/task-adaptive pretraining also improves transfer (Gururangan et al., 2020). Broader LLM data-efficiency work proposes LLM-guided quality scoring (Ask-LLM) and density sampling (Sachdeva et al., 2024), and clustering-based sensitivity sampling with provable guarantees (Axiotis et al., 2024). Finally, Data Filtering Networks (DFN) also leverage a held-out, high-quality set, but with a different goal and setup (Fang et al., 2023).

Our contribution differs by $(i)$ inverting train/validation roles to approximate per-example influence using only forward losses and doesn't require per-example gradients, or Hessian-vector products, and $(ii)$ showing that this simple, symmetry-based score is computationally inexpensive and outperforms recent data selection approaches for instruction tuning and NER.

## 2 DATA SELECTION FOR TOKEN-BASED LEARNING

In this section we describe our implementation of the general idea described in the introduction for token-based learning and present empirical results demonstrating its effectiveness. Since prediction takes place at the token level, while data selection takes place at the example level (e.g., instruction/output pair), we compute token scores and aggregate them as described in Section 2.1. Section 2.2 gives a brief overview of instruction-tuning and NER tasks. Experimental settings are introduced in Section 2.3. Empirical results are presented in Sections 2.4 and 2.5.

### 2.1 SCORE COMPUTATION FOR TOKEN-BASED LEARNING

Each example $z$ consists of an input $z^{\text{in}}$ and an output $z^{\text{out}}$, both of which are strings and may differ in length. Let $\mathcal{Z}^{\text{out}}$ denote the output vocabulary, and let $T(z)$ denote the length of the output string $z^{\text{out}}$, which we write as $z^{\text{out}} = \big(z^{\text{out}}(1), z^{\text{out}}(2), \ldots, z^{\text{out}}(T(z))\big)$.

Given a model parameterized by $\boldsymbol{\theta}$, its prediction on example $z$ is a sequence of $T(z)$ conditional distributions, $\{p_t(\cdot \mid z, \boldsymbol{\theta})\}_{t=1}^{T(z)}$, where each $p_t(\cdot \mid z, \boldsymbol{\theta})$ denotes the model's predictive distribution over the output token at position $t$. Note that $p_t(\cdot \mid z, \boldsymbol{\theta})$ depends on $z$ solely through $z^{\text{in}}$ and $z^{\text{out}}(1), \ldots, z^{\text{out}}(t-1)$. We train models using the log-loss

$$\ell(\boldsymbol{\theta}, z) = -\frac{1}{T(z)} \sum_{t=1}^{T(z)} \log p_t\big(z^{\text{out}}(t) \mid z; \boldsymbol{\theta}\big). \tag{7}$$

To compare two models, $\boldsymbol{\theta}$ and $\boldsymbol{\theta}'$, on example $\boldsymbol{z}$, we define a per-token difference of log-loss

$$\Delta_t(\boldsymbol{z}; \boldsymbol{\theta}, \boldsymbol{\theta}') = \log \frac{p_t\big(z^{\text{out}}(t) \mid \boldsymbol{z}; \boldsymbol{\theta}'\big)}{p_t\big(z^{\text{out}}(t) \mid \boldsymbol{z}; \boldsymbol{\theta}\big)}. \tag{8}$$

Since our setting involves selecting entire examples rather than individual tokens, we aggregate the per-token differences into a single score per example. Specifically, we apply a transformation function $F : \mathbb{R} \to \mathbb{R}$ to each $\Delta_t$ before averaging across positions. The final score for example $\boldsymbol{z}$ is:

$$\phi(\boldsymbol{z}; \boldsymbol{\theta}, \boldsymbol{\theta}') = \frac{1}{T(\boldsymbol{z})} \sum_{t=1}^{T(\boldsymbol{z})} F\big(\Delta_t(\boldsymbol{z}; \boldsymbol{\theta}, \boldsymbol{\theta}')\big). \tag{9}$$

We consider three instantiations of the function $F$, leading to three different scoring methods:

MAXIMUM-IMPROVEMENT: $F(y) = y$ — emphasizes raw improvement.

MAXIMUM-ABSOLUTE CHANGE: $F(y) = |y|$ — captures the magnitude of change.

MAXIMUM-POSITIVE IMPROVEMENT: $F(y) = \max\{y, 0\}$ — ignores degradations.

The algorithm is therefore the same as in Algorithm 1, with the adaptation $\phi_i^{(k)} = \phi(\boldsymbol{x}_i; \hat{\boldsymbol{\theta}}_k^{\text{bas}}, \hat{\boldsymbol{\theta}}_k^{\text{val}})$.

Given a budget of $n$ examples, we choose $S \subseteq [N] \setminus U$, $|S| = n$ using one of these rules:

SCORE-ONLY: Choose the $n$ examples $i \in [N] \setminus U$ that have the largest score $\phi_i$.

SCORE+RANDOM: Choose the $n/2$ examples $i \in [N] \setminus U$ that have the largest score $\phi_i$, and add $n/2$ more examples chosen uniformly at random (without replacement) from $U$.

Our scoring schemes tend to favor shorter examples due to their higher variance, which arises from having fewer tokens. To mitigate this bias, we partition the set $[N] \setminus U$ into 10 bins based on sequence length, ensuring each bin contains an equal number of examples. We then select an equal number of top-scoring examples from each bin.

After selecting $S$ of size $|S| = n$, we train (or fine tune) a model on $S$ to evaluate the selection scheme. We refer to this stage as *final training*.

We compare our schemes with three baselines:

RANDOM: The set $S$ is selected uniformly at random subject to its size.

MAXIMUM UNCERTAINTY: Instead of the scores we defined, we use the following hardness score:

$$\psi_i := \frac{1}{T_i} \sum_{t=1}^{T_i} \log \big(\mathsf{p}_t(z_i(t)|\boldsymbol{z}_i; \hat{\boldsymbol{\theta}}_L^{\text{bas}})(1 - \mathsf{p}_t(z_i(t)|\boldsymbol{z}_i; \hat{\boldsymbol{\theta}}_L^{\text{bas}}))\big), \tag{10}$$

This score extends the method of Ting & Brochu (2018); Wang et al. (2018); Ai et al. (2021); Kolossov et al. (2024) to token-based learning.

LESS: We used the publicly available implementation from Xia et al. (2024); see Appendix A.1.

## 2.2 PREDICTION TASKS

We evaluate our data selection framework in two distinct token-based tasks: instruction tuning (IT) and named entity recognition (NER). The framework we introduced above captures both tasks:

**Instruction Tuning (IT)** involves training a language model to follow natural language instructions. Each training example consists of:
**Input $\boldsymbol{z}^{\text{in}}$**: a user instruction or prompt; **Output $\boldsymbol{z}^{\text{out}}$**: the desired model response.

The output is typically multi-token and highly variable in content and length, depending on the instruction. The model learns to generate $\boldsymbol{z}^{\text{out}}$ conditioned on $\boldsymbol{z}^{\text{in}}$. This naturally fits our framework, which models predictions as token-level distributions $p_t(\cdot \mid \boldsymbol{z}, \boldsymbol{\theta})$.

**Named Entity Recognition (NER)** is a sequence labeling task where the model assigns a probability distribution over entity tags (e.g., PERSON, ORGANIZATION, . . . ) to each token. In this case:
**Input $\boldsymbol{z}^{\text{in}}$**: a tokenized sentence; **Output $\boldsymbol{z}^{\text{out}}$**: a sequence of entity labels, aligned with the input.

In NER, predictions are computed as token-wise classification distributions and therefore output is of the same length as input sequence.[1] In this case, as a base model we take a pretrained language model and replace its prediction head with a classification head.

## 2.3 EXPERIMENTAL SETTING

In all of our experiments the training set consisted of $N = 36 \times 1024$ samples. For the base model training, we used $|U| = 4 \times 1024$ samples. The validation set size is $m_{\text{val}} = 1024$ and the test set size is $m_{\text{tst}} = 10,000$. We vary the selected set size $n \in \{1, 2, 4, 8\} \times 1024$.

**Number of epochs.** Both for surrogate model training and final model training we determine the number of epochs by $L = (16 \times 1024)/n_{\text{tr}}$. We use a batch size of 16 whence the above ensures that the number batches used in training remains constant, and equal to 1024. In other words, all experiments in this section are at *constant compute*. Since base model training uses $|U| = m = 4 \times 1024$ samples, the number of epochs is $L = 4$.

**Learning rate.** The learning rate for both surrogate and final model training is selected using hyper-parameter optimization for each selected set size $n$. The learning-rate optimization was carried out for random data selection hence placing our approach at a disadvantage.

We use linear learning rate scheduler and LoRA training Hu et al. (2022) with LoRA parameters $\alpha = 32$ and dropout $= 0.2$. For NER experiments, we used PEFTrank $= 1$ and for instruction tuning experiments, we used PEFTrank $= 256$. The learning rate for the validation examples is $\varepsilon = 1/10$ of the one for the base examples. We present here results with SCORE+RANDOM and refer to the appendix for SCORE-ONLY.

## 2.4 EXPERIMENTS FOR INSTRUCTION TUNING

For these experiments we used 3 different datasets, which we will refer to as $\mathscr{S} := \{$Slim Orca, Alpaca GPT-4, Alpaca GPT-3.5$\}$. As the foundation model, we use **Meta-Llama-3-8B**. Additional details of the model and datasets used are provided in the Appendix.

We designed five experimental setups. In each experiment, one dataset from $\mathscr{S}$ is selected as the *target distribution*. We randomly sample validation and test sets, $\mathbf{Z}^{\text{val}}$ and $\mathbf{Z}^{\text{tst}}$, without replacement from the target dataset. These samples are excluded from further use. The *training pool* is then formed by randomly sampling an equal number of examples from one or more datasets in $\mathscr{S}$ (excluding the validation and test samples), such that the total number of selected training samples is fixed at $N$. We denote by $\mathscr{S}_* \subseteq \mathscr{S}$ the datasets used to generate the training pool. The choices of the target dataset and of $\mathscr{S}_*$ for each of the five experiments are summarized in Table 1. All reported results are averaged over 10 independent runs. In each run, we freshly sample the training, validation, and test sets. These experiments are designed to evaluate performance across a range of data configurations. In particular: in Experiments 1 and 4, the training set includes samples from both target distribution and other distributions; in Experiments 2 and 5, the training set includes samples only from non-target distributions; in Experiment 3, it includes only samples from the target distribution.

Figure 1 summarizes our results for instruction tuning for a fixed select size $n = 8 \times 1024$. We plot the improvement in test log-loss over random data selection for several data-selection strategies within the general framework described in Section 2.1, using Interleaved ToV scoring in algorithm 1 for scoring the examples and SCORE+RANDOM for selecting. We observe that the proposed strategies yield significantly better instruction tuning than random data selection or selecting by max-uncertainty. We observe an improvement (albeit a small one) even when both train and validation data are from Slim Orca (Exp 3), which is a case in which random selection should perform well. The proposed strategies also yield a significant improvement over LESS (Xia et al., 2024), with the exception of Experiment 2 in which LESS performs slightly better.

Figure 2 displays the evolution of test log loss with selected sample size $n$. We observe that a good choice of the data selection method results in model improvements that can be equivalent to or larger than doubling $n$. Plots show standard error (with scaling factor 1) for 10 runs.

---

[1] In NER, typically token level probabilities are combined to assign labels to a whole word.

Table 1: Summary of instruction tuning experiments. Abbreviations: SO = Slim Orca, A4 = Alpaca GPT-4, A3.5 = Alpaca GPT-3.5.

| Exp | Target | Training pool |
|-----|--------|---------------|
| 1 | SO | SO, A4, and A3.5 |
| 2 | SO | A4 and A3.5 |
| 3 | SO | SO |
| 4 | A4 | SO, A4, and A3.5 |
| 5 | A4 | SO and A3.5 |

Table 2: Summary of named entity recognition experiments. Abbreviations: MN = Multinerd, A4p = Ai4p, C4 = C4, SB = Syn-big.

| Exp | Target | Training pool |
|-----|--------|---------------|
| 1 | MN | MN, A4p, C4, and SB |
| 2 | MN | A4p, C4, and SB |
| 3 | MN | MN |
| 4 | A4p | MN, A4p, C4, and SB |
| 5 | A4p | MN, C4, and SB |
| 6 | A4p | A4p |

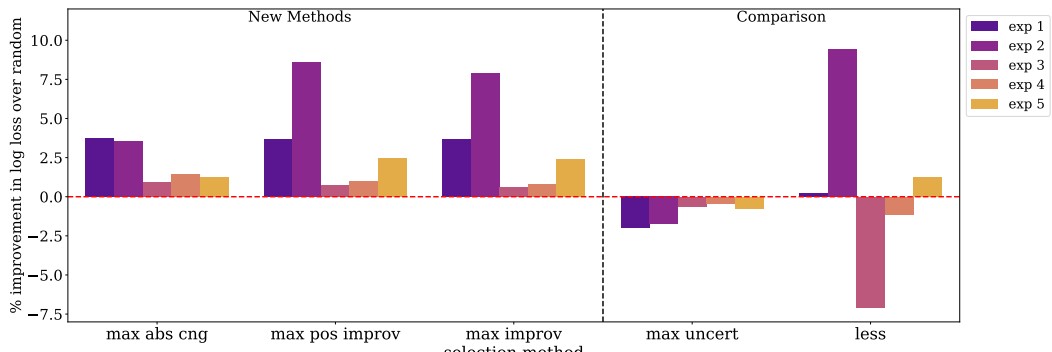

Figure 1: Test log-loss improvement (%) over random selection for instruction tuning with $n = 8 \times 1024$ samples. Each group of bars represents a data-selection strategy (maximum-uncertainty and LESS as baselines); colors show target/training pool configuration (Table 1). Results use Interleaved ToV scoring (Algorithm 1) with the SCORE+RANDOM selection strategy.

## 2.5 EXPERIMENTS FOR NAMED ENTITY RECOGNITION

The task is to classify whether a token is part of a person name or not. For these experiments we used 4 different labeled datasets, which we will refer to as $\mathscr{S} := \{$ Multinerd, Ai4p, C4, Syn-big$\}$. We use **xlm-roberta-base** as the foundation model. Further details on the experiment, model and datasets used are presented in the Appendix.

We conducted six sets of experiments. As for the case of instruction tuning, for each set of experiments, we select one of the datasets $\mathscr{S}$ as defining the target distribution, and one or more other datasets to define the training pool (denoted by $\mathscr{S}_*$). The choices of target datasets and $\mathscr{S}_*$ are summarized in Table 2. The construction of train, test and validation sets is same as in instruction tuning.

Figure 3 summarizes our experiments with NER. We plot the improvement in test log-loss over random data selection for several score definitions. Throughout these experiments, we use SCORE+RANDOM. We observe that the strategies of Section 2.1 yield systematic improvements over random data-selection. Unlike in instruction tuning, maximum uncertainty also improves performance in most settings; however, ToV achieves the largest gains. In this case, LESS (Xia et al., 2024) does not improve over random selection.

We also evaluated token-level F1 on the NER task and observed trends consistent with the log-loss improvements across selection methods (App. A.4).

## 2.6 RUNTIME AND MEMORY.

We compared the computational cost of ToV and LESS under identical hardware and software settings (Appendix K.3) using the official LESS implementation. We report wall-clock runtime and

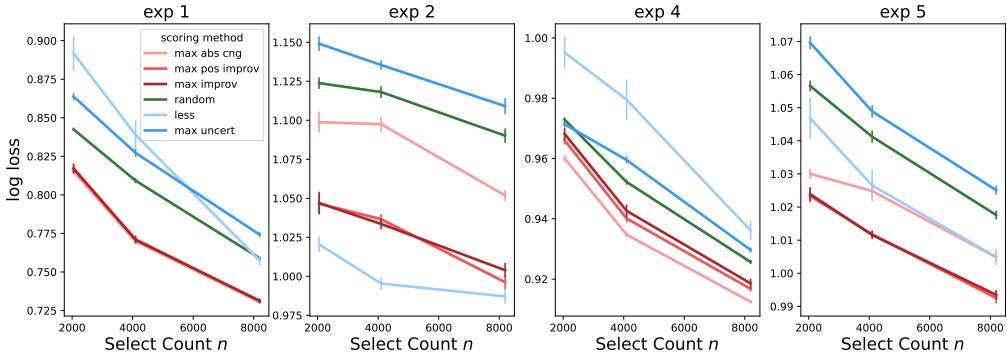

Figure 2: Test log-loss vs. number of selected samples $n$ for instruction tuning. (Due to space limits, Exp. 3 plot is in the Appendix.) Lines show mean log-loss over 10 runs; error bars are $\pm 1$ standard error. Results use Interleaved ToV scoring with the SCORE+RANDOM strategy.

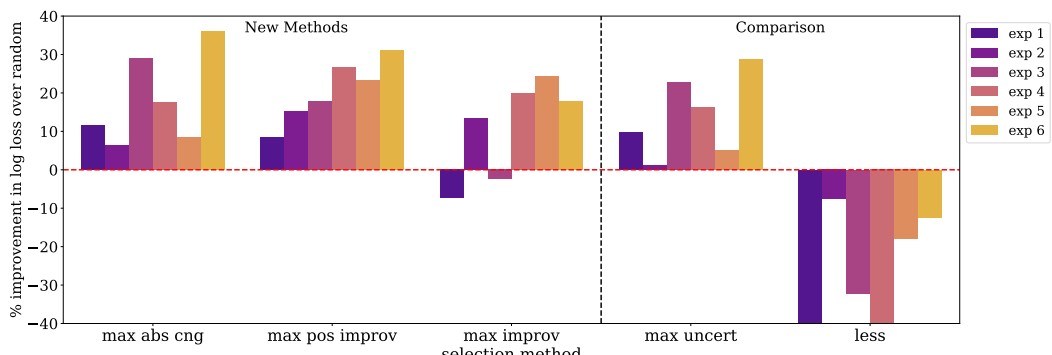

Figure 3: Test log-loss improvement (%) relative to random selection for NER at $n = 8 \times 1024$. Each group of bars represents a data-selection strategy; colors show target/training pool configuration (Table 2). Results use Interleaved ToV scoring (Algorithm 1) with the SCORE+RANDOM strategy

disk storage required by each method (e.g., checkpoints and auxiliary tensors). Reported values are averages over five runs per task. Across both instruction tuning and NER, ToV is substantially faster and more storage-efficient. Results are summarized in Table 3.

Table 3: Runtime and disk storage for ToV vs. LESS under identical hardware (mean of 5 runs).

| Setting | Method | Runtime | Storage |
|---|---|---|---|
| Instruction tuning | LESS | 4h 5m | 4.9 GB |
| Instruction tuning | ToV | 2h 3m | 1.84 GB |
| NER | LESS | 46m | 4.1 GB |
| NER | ToV | 8m | 0.24 GB |

ToV reduces runtime by 2–6$\times$ and disk storage by 2.5–16$\times$ relative to LESS across the evaluated tasks.

## 3 A FORMAL JUSTIFICATION

In this section, we present a mathematical analysis of our approach in the case of batch gradient descent (GD). We focus on the implementation *Parallel ToV* (Method B), described in Algorithm 2.

Parallel ToV differs from Interleaved ToV because at each training cycle $k$, training on the base set $\boldsymbol{X}_U$ is initialized with the output of the previous train-on-validation phase. Empirically Interleaved

---

**Algorithm 2** ToV Scoring Algorithm: *Parallel ToV*

---

1: **Input:** Pretrained model $\boldsymbol{\theta}_0$, validation set $\boldsymbol{Z}^{\text{val}}$, training pool $\boldsymbol{X} = (\boldsymbol{x}_i : i \in [N])$,
2:         selected data count $n \le N$, base model count $m$
3: **Output:** Set of examples $S \subset [N]$ of size $n$
4: Sample base subset $U \subseteq [N]$ of size $m$ randomly; define $\boldsymbol{X}_U = (\boldsymbol{x}_i : i \in U)$
5: Initialize models: $\hat{\boldsymbol{\theta}}_0^{\text{bas},+} \leftarrow \boldsymbol{\theta}_0$, $\hat{\boldsymbol{\theta}}_0^{\text{bas}} \leftarrow \boldsymbol{\theta}_0$; set scores $\Upsilon_i \leftarrow 0$ for all $i \in [N] \setminus U$
6: **for** $k = 1$ to $L$ **do**
7:      Train for one epoch on $\boldsymbol{X}_U$ with learn. rate $\eta_k$ and init. $\hat{\boldsymbol{\theta}}_{k-1}^{\text{bas},+}$. Denote the output by $\hat{\boldsymbol{\theta}}_{0,k}^{\text{bas},+}$
8:      Train for one epoch on $\boldsymbol{Z}^{\text{val}}$ with learn. rate $\varepsilon \cdot \eta_k$ and init. $\hat{\boldsymbol{\theta}}_{0,k}^{\text{bas},+}$. Denote the output by $\hat{\boldsymbol{\theta}}_k^{\text{bas},+}$
9:      Train for one epoch on $\boldsymbol{X}_U$ with learn. rate $\eta_k$ and init. $\hat{\boldsymbol{\theta}}_{k-1}^{\text{bas}}$. Denote the output by $\hat{\boldsymbol{\theta}}_k^{\text{bas}}$
10:      **for** each $i \in [N] \setminus U$ **do**
11:         $\Upsilon_i^{(k)} \leftarrow \ell(\hat{\boldsymbol{\theta}}_k^{\text{bas}}; \boldsymbol{x}_i) - \ell(\hat{\boldsymbol{\theta}}_k^{\text{bas},+}; \boldsymbol{x}_i)$
12:         $\Upsilon_i \leftarrow \Upsilon_i + \Upsilon_i^{(k)}/L$
13:      **end for**
14: **end for**
15: Select $S \subseteq [N] \setminus U$ with size $|S| = n$ using scores $\Upsilon_i$

---

ToV performs somewhat better than Parallel ToV, see Appendix C. We use Parallel ToV for analysis just because the resulting mathematical expressions are simpler.

We find empirically that the ToV works well beyond token-based learning, and hence our focus will be to understand it in a generic learning problem. Appendix D demonstrates this point by considering a simple logistic regression problem.

## 3.1   Ideal scores, linearization, influence functions

In order to estimate the model improvement produced by sample $i \in [N] \setminus U$ we could train a model on two training sets $\boldsymbol{X}_U$ and $\boldsymbol{X}_{U \cup i}$, using empirical risk functions $\widehat{R}_U(\boldsymbol{\theta})$, $\widehat{R}_{U \cup i}(\boldsymbol{\theta})$. We thus would run GD for $L$ steps, with initialization $\hat{\boldsymbol{\theta}}_0^{\text{bas}} = \hat{\boldsymbol{\theta}}_0^{\text{bas}+i} = \boldsymbol{\theta}_0$:

$$\hat{\boldsymbol{\theta}}_{k+1}^{\text{bas}} = \hat{\boldsymbol{\theta}}_k^{\text{bas}} - \eta m \nabla \widehat{R}_U(\hat{\boldsymbol{\theta}}_k^{\text{bas}}), \quad \hat{\boldsymbol{\theta}}_{k+1}^{\text{bas}+i} = \hat{\boldsymbol{\theta}}_k^{\text{bas}+i} - \eta(m+1)\nabla \widehat{R}_{U \cup i}(\hat{\boldsymbol{\theta}}_k^{\text{bas}+i}). \tag{11}$$

At iteration $k$, we have thus two models $\hat{\boldsymbol{\theta}}_k^{\text{bas}}$ and $\hat{\boldsymbol{\theta}}_k^{\text{bas}+i}$ that differ uniquely in whether sample $i$ is used or not. We define the *ideal score* to be the difference in validation error between these two models, averaged over epochs

$$S_i := \frac{1}{L} \sum_{s=1}^{L} [\widehat{R}_{\text{val}}(\hat{\boldsymbol{\theta}}_s^{\text{bas}}) - \widehat{R}_{\text{val}}(\hat{\boldsymbol{\theta}}_s^{\text{bas}+i})] = \frac{1}{m_{\text{val}} L} \sum_{s=1}^{L} \sum_{j=1}^{m_{\text{val}}} \left\{ \ell(\hat{\boldsymbol{\theta}}_s^{\text{bas}}; \boldsymbol{z}_j^{\text{val}}) - \ell(\hat{\boldsymbol{\theta}}_s^{\text{bas}+i}; \boldsymbol{z}_j^{\text{val}}) \right\}. \tag{12}$$

Evaluating this score is computationally expensive, hence several groups (Pruthi et al., 2020; Bae et al., 2024; Xia et al., 2024) proposed to use a first order Taylor expansion to approximate the difference between the two models. Expanding $S_i$ with respect to the contribution of $\ell(\cdot; \boldsymbol{x}_i)$ yields

$$S_i^{\text{lin}} = \frac{\eta}{L} \sum_{0 \le s < t \le L} \langle \nabla \widehat{R}_{\text{val}}(\hat{\boldsymbol{\theta}}_t^{\text{bas}}), \boldsymbol{M}_{t,s+1} \nabla \ell(\hat{\boldsymbol{\theta}}_s^{\text{bas}}; \boldsymbol{x}_i) \rangle. \tag{13}$$

where $\boldsymbol{M}_{t,t} = \boldsymbol{I}_d$ and $\boldsymbol{M}_{t,r}$ captures the propagation of perturbations along the GD trajectory:

$$\boldsymbol{M}_{t,r} := \boldsymbol{H}_{t-1} \cdot \boldsymbol{H}_{t-2} \cdots \boldsymbol{H}_r, \qquad \boldsymbol{H}_k := \boldsymbol{I} - \eta m \nabla^2 \widehat{R}_U(\hat{\boldsymbol{\theta}}_k^{\text{bas}}). \tag{14}$$

The next result shows that $S_i^{\text{lin}}$ approximates well $S_i$ in a quantitative way, under local convexity.

**Proposition 1.** *Assume there exist $c_0, C_1, M > 0$ such that $\nabla^2 \widehat{R}_U(\hat{\boldsymbol{\theta}}_k^{\text{bas}}) \succeq c_0 \boldsymbol{I}_d$, $\|\nabla \ell(\hat{\boldsymbol{\theta}}_k^{\text{bas}}; \boldsymbol{x}_i)\| \le C_1$ for all $k$ and, for all $\boldsymbol{\theta}_1, \boldsymbol{\theta}_2$, $\|\nabla^2 \widehat{R}_U(\boldsymbol{\theta}_1) - \nabla^2 \widehat{R}_U(\boldsymbol{\theta}_2)\|_{\text{op}} \le M \|\boldsymbol{\theta}_1 - \boldsymbol{\theta}_2\|_2$, $\|\nabla \ell(\boldsymbol{\theta}_1; \boldsymbol{x}_i) - \nabla \ell(\boldsymbol{\theta}_2; \boldsymbol{x}_i)\|_{\text{op}} \le M \|\boldsymbol{\theta}_1 - \boldsymbol{\theta}_2\|_2$. Further assume that $\|\nabla^2 \widehat{R}_{\text{val}}(\hat{\boldsymbol{\theta}}_k^{\text{bas}})\|_{\text{op}} \le C_1$ and $\|\nabla^2 \widehat{R}_{\text{val}}(\boldsymbol{\theta}_1) - \nabla^2 \widehat{R}_{\text{val}}(\boldsymbol{\theta}_2)\|_{\text{op}} \le M \|\boldsymbol{\theta}_1 - \boldsymbol{\theta}_2\|_2$ for all $\boldsymbol{\theta}_1, \boldsymbol{\theta}_2$ as well. Finally, assume there exists a constant $C_\eta$ such that $\eta_k = \eta \le C_\eta/m$ $\forall k$. Then there exists $C = C(c_0, C_1, C_\eta, M)$ such that*

$$\left| S_i - S_i^{\text{lin}} \right| \le C/m^2. \tag{15}$$

The assumption $\eta \leq C_\eta/m$ is justified by the fact that we expect the Hessian of $\widehat{R}_U(\cdot)$ to be of order one, and hence the stepsize for this objective (which is given by $\eta m$ see Eq. (11)) should be of order one. As shown in the proof, the typical size of $S_i^{\text{lin}}$ is of order $1/m$, and hence Eq. (15) establishes that the difference $|S_i - S_i^{\text{lin}}|$ is negligible.

We emphasize that Proposition 1 does not make any assumption about the data distribution and instead entirely relies on regularity properties of the risk function.

## 3.2 TRAIN-VALIDATION DUALITY

We consider Interleaved ToV and Parallel ToV defined in Algorithms 1, 2. We emphasize the dependence on $\varepsilon$ by writing $\phi_i = \phi_i(\varepsilon)$ and $\Upsilon_i = \Upsilon_i(\varepsilon)$. It is easy to derive the small $\varepsilon$ asymptotics $\phi_i = \phi_i^{\text{lin}}\varepsilon + o(\varepsilon)$, $\Upsilon_i = \Upsilon_i^{\text{lin}}\varepsilon + o(\varepsilon)$, where, for $\boldsymbol{g}_{s,i} := \nabla\ell(\hat{\boldsymbol{\theta}}_s^{\text{bas}}; \boldsymbol{x}_i)$,

$$\phi_i^{\text{lin}} := \frac{\eta m_{\text{val}}}{L}\sum_{s=1}^{L}\langle\nabla\widehat{R}_{\text{val}}(\hat{\boldsymbol{\theta}}_s^{\text{bas}}), \boldsymbol{g}_{s,i}\rangle, \qquad \Upsilon_i^{\text{lin}} := \frac{\eta m_{\text{val}}}{L}\sum_{0\leq t < s \leq L}\langle\nabla\widehat{R}_{\text{val}}(\hat{\boldsymbol{\theta}}_{t+1}^{\text{bas}}), \boldsymbol{M}_{s,t+1}^{\mathsf{T}}\boldsymbol{g}_{s,i}\rangle. \quad (16)$$

We show that these are good approximations of $\Upsilon_i(\varepsilon), \phi_i(\varepsilon)$ uniformly in dimension, sample size.

**Theorem 1.** *Consider Algorithms 1, 2 with fixed stepsize $\eta_k = \eta$ (and $F(x) = -x$ in Algorithm 1). Under the assumptions of Proposition 1, further assume $\|\nabla\widehat{R}_{\text{val}}(\hat{\boldsymbol{\theta}}_k^{\text{bas}})\| \leq C_1$ for all $k$, and $\|\nabla_{\boldsymbol{\theta}}^2\ell(\boldsymbol{\theta}; \boldsymbol{x})\|_{\text{op}} \leq C_1$. Then there exist $c_* = c_*(c_0, M, C_1)$, $C = C(c_0, M, C_1)$ such that, for $\varepsilon m_{\text{val}}/m \leq c_*$,*

$$\left|\Upsilon_i(\varepsilon) - \Upsilon_i^{\text{lin}}\varepsilon\right| \leq C\left(\varepsilon m_{\text{val}}/m\right)^2, \quad \left|\phi_i(\varepsilon) - \phi_i^{\text{lin}}\varepsilon\right| \leq C\left(\varepsilon m_{\text{val}}/m\right)^2. \quad (17)$$

Note that $\Upsilon_i^{\text{lin}}$ differ from $S_i^{\text{lin}}$. because of: $(i)$ The different order of $s$ and $t$; $(ii)$ The fact that $\boldsymbol{M}_{t,s+1}$ is replaced by its transpose in Eq. (16). $\Upsilon_i^{\text{lin}}$ measures the influence of training on validation data when making inference at $\boldsymbol{x}_i$, while $S_i^{\text{lin}}$ measures the influence of training on $\boldsymbol{x}_i$ data when making inference on validation. These two measures of 'influence' differ by the replacement of $\boldsymbol{M}_{t,s+1}$ by $\boldsymbol{M}_{s,t}^{\mathsf{T}}$. However, in a number of cases we expect these two matrices to be not too different, and hence the two scores to yield similar results. We can prove that $\Upsilon_i^{\text{lin}}$ and $S_i^{\text{lin}}$ coincide (for large $L$) under local convexity conditions.

**Theorem 2.** *Assume $\boldsymbol{\theta} \mapsto \ell(\boldsymbol{\theta}; \boldsymbol{x})$ to be twice continuously differentiable and that $\|\nabla\widehat{R}_{\text{val}}(\hat{\boldsymbol{\theta}}_k^{\text{bas}})\| \leq C_1$, $\|\nabla\ell(\hat{\boldsymbol{\theta}}_k^{\text{bas}}; \boldsymbol{x}_i)\| \leq C_1$ for all $k$. Further assume that gradient descent iterates $(\hat{\boldsymbol{\theta}}_k^{\text{bas}} : k \geq 0)$ converge to $\hat{\boldsymbol{\theta}}_\infty^{\text{bas}} = \lim_{k\to\infty}\hat{\boldsymbol{\theta}}_k^{\text{bas}}$ which is a local minimum of $\widehat{R}_U(\boldsymbol{\theta})$ with $\boldsymbol{Q}_\infty := \nabla^2\widehat{R}_U(\hat{\boldsymbol{\theta}}_\infty^{\text{bas}}) \succ \boldsymbol{0}$ (strictly). Then*

$$\lim_{L\to\infty}\frac{1}{m_{\text{val}}}\Upsilon_i^{\text{lin}}(L) = \lim_{L\to\infty}S_i^{\text{lin}}(L) = \frac{1}{m}\langle\nabla\widehat{R}_{\text{val}}(\hat{\boldsymbol{\theta}}_\infty^{\text{bas}}), \boldsymbol{Q}_\infty^{-1}\nabla\ell(\hat{\boldsymbol{\theta}}_\infty^{\text{bas}}; \boldsymbol{x}_i)\rangle := S_{i,\infty}^{\text{lin}}. \quad (18)$$

The last expression in Eq. (18) (denoted by $S_{i,\infty}^{\text{lin}}$) is the classical formula for influence functions of M-estimators (van der Vaart, 2000). Both our approach and the dynamical influence function $S_i^{\text{lin}}(L)$ can be regarded as approximations of $S_{i,\infty}^{\text{lin}}$ in this case.

In fine tuning, the model is likely to be overparametrized, and it is unrealistic to assume convergence to a strict minimum (with $\nabla^2\widehat{R}_U(\hat{\boldsymbol{\theta}}_\infty^{\text{bas}}) \succ \boldsymbol{0}$). On the other hand, the weights will not change significantly during this phase and it is reasonable to approximate fine-tuning as fitting an overparametrized linear model with respect to the empirical neural tangent features learnt in the pre-training phase.

**Theorem 3.** *Consider the loss function $\ell(\boldsymbol{\theta}; \boldsymbol{x}) = (y(\boldsymbol{x}) - \langle\boldsymbol{\psi}(\boldsymbol{x}), \boldsymbol{\theta}\rangle)^2/2$ for some response variables $y(\boldsymbol{x})$, and featurization map $\boldsymbol{\psi} : \mathbb{R}^d \to \mathbb{R}^p$, $p > m$. Let $\boldsymbol{\Psi} \in \mathbb{R}^{|U|\times p}$ be the matrix with rows $(\boldsymbol{\psi}(\boldsymbol{x}_j) : j \in U)$, $\boldsymbol{\Psi}_{\text{val}} \in \mathbb{R}^{m_{\text{val}}\times p}$ be the matrix with rows $(\boldsymbol{\psi}(\boldsymbol{z}_j^{\text{val}}) : j \leq m_{\text{val}})$, $\boldsymbol{P}_\Psi$ the projector to the kernel of $\boldsymbol{\Psi}$, $\boldsymbol{y} = (y(\boldsymbol{x}_j) : j \in U)$, $\hat{\boldsymbol{\theta}} := \boldsymbol{\Psi}^\dagger\boldsymbol{y}$, $\boldsymbol{r}^{\text{val}} := (y(\boldsymbol{z}_j^{\text{val}}) - \langle\hat{\boldsymbol{\theta}}, \boldsymbol{\psi}(\boldsymbol{z}_j^{\text{val}})\rangle : j \leq m_{\text{val}})$, $r(i) := y(\boldsymbol{x}_i) - \langle\hat{\boldsymbol{\theta}}, \boldsymbol{\psi}(\boldsymbol{x}_i)\rangle$. If GD is initialized with $\boldsymbol{\theta}_0 = \boldsymbol{0}$, and we use constant stepsize $\eta < \|\boldsymbol{\Psi}\|_{op}^2/2$, then*

$$\lim_{L\to\infty}\frac{1}{Lm_{\text{val}}}\Upsilon_i^{\text{lin}}(L) = \lim_{L\to\infty}\frac{1}{L}S_i^{\text{lin}}(L) = \frac{\eta}{2}r(i)\langle\boldsymbol{r}^{\text{val}}, \boldsymbol{\Psi}_{\text{val}}^{\mathsf{T}}\boldsymbol{P}_\Psi\boldsymbol{\psi}(\boldsymbol{x}_i)\rangle. \quad (19)$$

## ACKNOWLEDGEMENTS

We are grateful to Neeraja Abhyankar, Alankrita Bhatt, Joseph Gardi, Mukur Gupta, Germain Kolossov, Marc Laugharn, Rahul Ponnala, Sahasrajit Sarmasarkar, Andreas Santucci and Pulkit Tandon, for several conversations about this work.

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

## A  ADDITIONAL EXPERIMENTAL DETAILS AND RESULTS

### A.1  EXPERIMENTS FOR TOKEN-BASED LEARNING

In these experiments, we used pretrained models as base models and constructed training, validation, and test sets from real-world datasets. Details of the datasets and models are provided in Section K.

For each training example count—both for surrogate model training (used for scoring) and final model training—we selected the learning rate from the following grid:

$$[3e\text{-}6,\ 1e\text{-}5,\ 3e\text{-}5,\ 1e\text{-}4,\ 3e\text{-}4,\ 1e\text{-}3,\ 3e\text{-}3,\ 1e\text{-}2].$$

The optimal learning rate was determined by training models on randomly sampled subsets from the training pool and evaluating their test log-loss. For each learning rate, the loss was averaged over 10 runs, with a new random subset used in each run. The best-performing learning rate was selected separately for each experimental configuration listed in Table 1 and Table 2.

**Implementation details for Less (Xia et al., 2024)** We used the public implementation from the authors' GitHub repository. The projection dimension was set to 8192. Learning rate and other hyperparameters were tuned identically for all approaches. For both our method and LESS, the surrogate model used the same number of samples and was trained for four epochs, matching the settings in the LESS paper. Following the original LESS procedure, we selected the top-scoring examples.

### A.2  EXPANDED RESULTS FOR INSTRUCTION TUNING

In the main paper, we compared our scoring methods for the SCORE+RANDOM strategy. Due to space constraints, Figure 2 omitted results for Experiment 3. In Figure 4, we provide an expanded version that includes results for Experiment 3 as well.

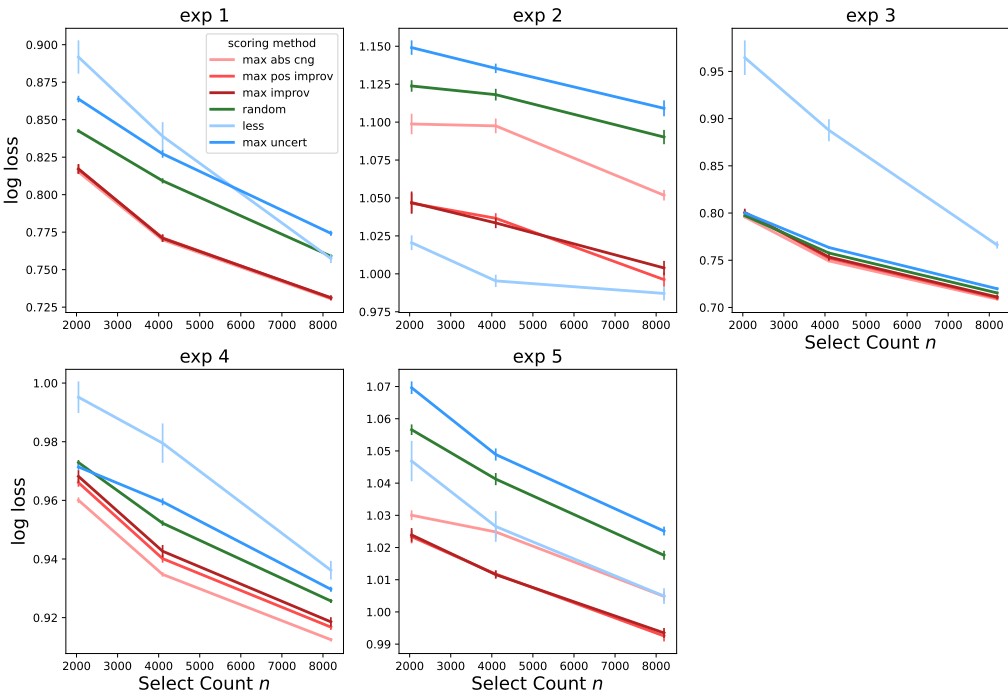

Figure 4: Expanded version of Figure 2 including the Experiment 3 plot.

### A.3 EXPANDED RESULTS FOR NAMED ENTITY RECOGNITION

In Figure 3 of the main paper, we reported results for SCORE+RANDOM using a fixed selected sample size of $n = 8 \times 1024$, across all experiment configurations in Table 2.

In Figure 5, we show how the test log-loss varies with the selected sample size $n$ for different scoring methods under the SCORE+RANDOM strategy, and how these compare to random selection.

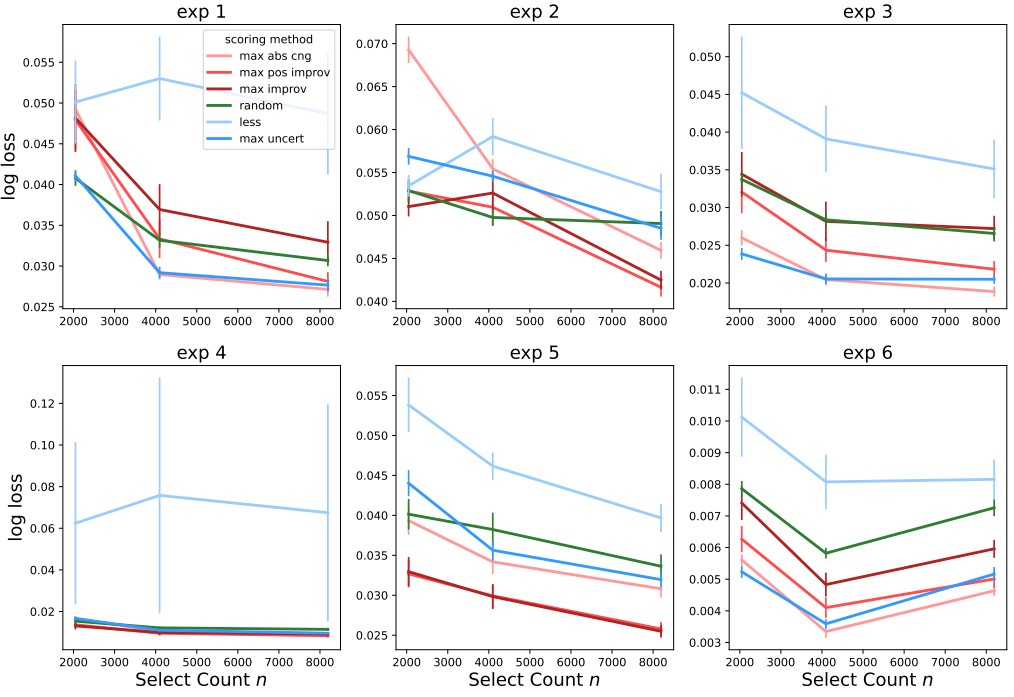

Figure 5: Test log-loss vs. number of selected samples $n$ for NER. Lines show mean log-loss over 10 runs; error bars are $\pm 1$ standard error. Results use Interleaved ToV with the SCORE+RANDOM strategy.

### A.4 ADDITIONAL NER RESULTS: TOKEN-LEVEL F1

To complement the log-loss evaluation reported above and in the main paper, we also measure token-level F1 on the NER task. Figure 6 shows test $1 - \text{F1}$ as a function of the number of selected samples $n$, using the same experimental setup as Fig. 5. Lower values therefore, indicate better sequence labeling performance.

We observe trends consistent with the log-loss results: our selection methods outperform random selection and competing baselines in several settings, confirming that cross-entropy gains translate into improved token-level F1.

## B COMPARISON OF SCORE+RANDOM AND SCORE-ONLY SELECTION

In this section, we examine how the performance of our strategies changes when all training examples are selected from the top-scoring set (SCORE-ONLY) instead of selecting only half of them from the top and the other half at random (SCORE+RANDOM).

Recall that our scores approximate how much benefit each example provides when added to a randomly chosen pool of training data. A higher score therefore indicates an example expected to be

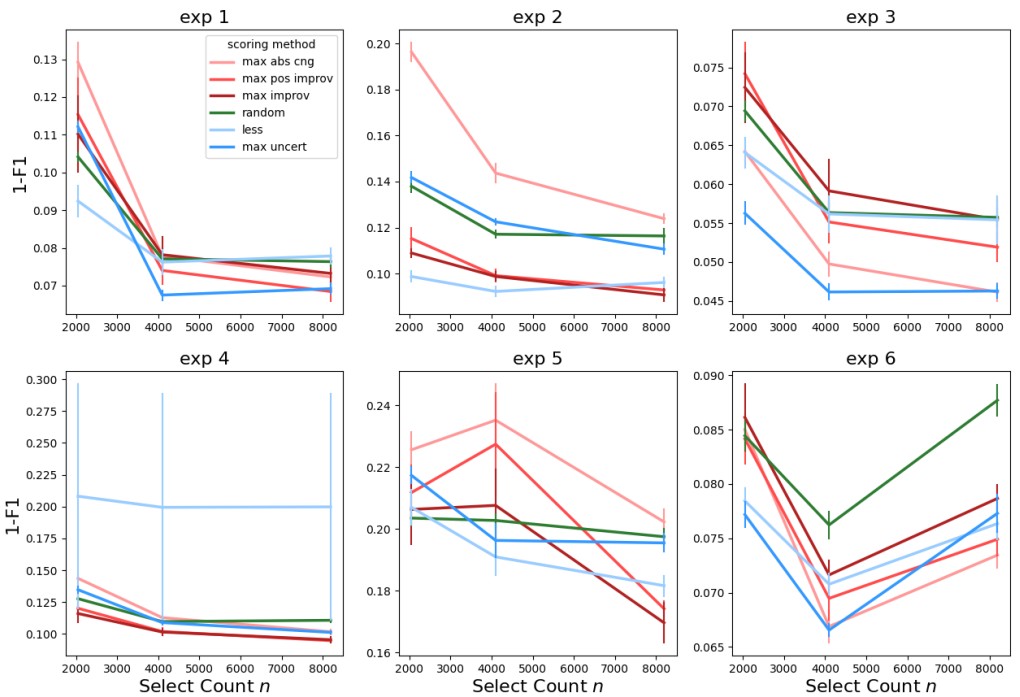

Figure 6: Test $1 - F1$ vs. number of selected samples $n$ for NER. Lines show mean $1 - F1$ over 10 runs; error bars denote $\pm 1$ standard error. Results use Interleaved ToV with the SCORE+RANDOM strategy.

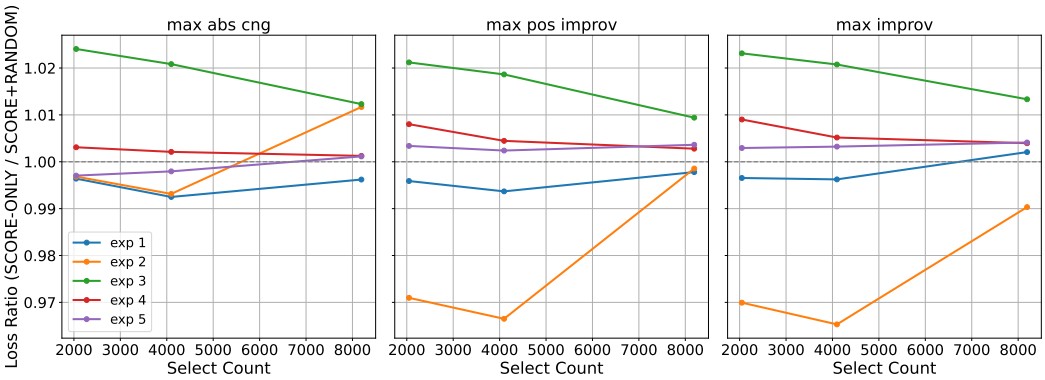

Figure 7: Ratio of log loss for SCORE-ONLY versus SCORE+RANDOM across our three scoring strategies and all instruction-tuning setups in Table 1. Scores are computed using Algorithm 1.

more helpful in that setting. SCORE+RANDOM selects half of the final training set from the highest-scoring examples and fills the rest with random examples, whereas SCORE-ONLY takes only the top-scoring examples. This design creates a trade-off:

- Pure exploitation: Selecting only top-scoring examples can maximize immediate gain because every chosen example has a high estimated contribution.
- Score validity and diversity: The scores are defined relative to adding examples to a random pool. If we select only top examples, the resulting set may differ substantially from the

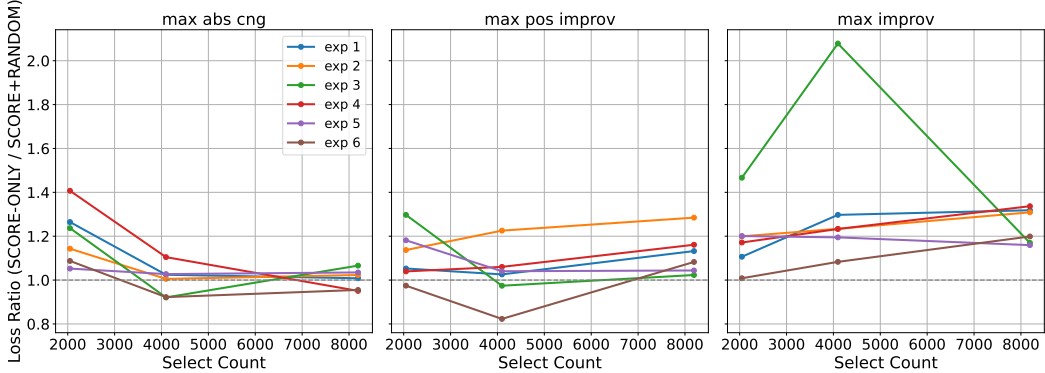

Figure 8: Ratio of log-loss for SCORE-ONLY versus SCORE+RANDOM across our three scoring strategies and all NER setups in Table 2. Scores are computed using Algorithm 1.

random reference, making the scores a less accurate guide. Randomly adding half the examples keeps the final set closer to the conditions under which the scores were computed and also protects against loss of diversity.

Which effect dominates varies by task.

Figures 7 and 8 show the ratio of log-loss for the two selection strategies in instruction tuning and NER respectively. In each figure the three subplots correspond to our three scoring strategies; different lines indicate the various experimental setups. Algorithm 1 is used to obtain the scores.

For instruction tuning, SCORE+RANDOM performs better in three setups (3, 4, 5), while SCORE-ONLY is better in the remaining two (1, 2) across most selection sizes and scoring methods. For NER, SCORE+RANDOM tends to outperform SCORE-ONLY more often, particularly for the Max-Improvement scores.

## C  PARALLEL TOV VS INTERLEAVED TOV

All previous plots used Interleaved ToV (Algorithm 1) for scoring. Here we compare the performance of the two scoring methods: Interleaved ToV (Algorithm 1) and Parallel ToV (Algorithm 2)—across our experiments, using the SCORE+RANDOM selection strategy for both.

Figures 9 and 10 show the ratio of test log-loss obtained with Parallel ToV relative to Interleaved ToV for instruction tuning and NER, respectively. Each figure contains three subplots corresponding to our three scoring strategies, and different lines represent the various experimental setups.

The results indicate that for instruction tuning, Interleaved ToV is most often superior, while for NER there is no consistent winner. A possible explanation is that Parallel ToV uses two distinct training trajectories. Our analysis assumes that the resulting models differ only slightly, but in practice, the two training trajectories can diverge substantially. This effect is likely to be stronger with large and highly overparameterized models such as Meta-Llama-3-8B, which we used for instruction tuning, We expect the larger distance between the two models to result in less accurate score estimation in Parallel ToV, as compared to Interleaved ToV.

## D  LOGISTIC REGRESSION EXPERIMENTS

In these experiments, we synthetically generated the training pool, validation set, and test set. We begin by defining a parametric family of distributions used to construct the data.

For a given $p > 0$ and parameter vector $\theta \in \mathbb{R}^p$, we define a distribution $\mathcal{P}_\theta$ over pairs $(x, y)$, where $x \in \mathbb{R}^p$ and $y \in \{0, 1\}$. The features are sampled as $x \sim \mathcal{N}(0, I)$, and the label $y$ is drawn according

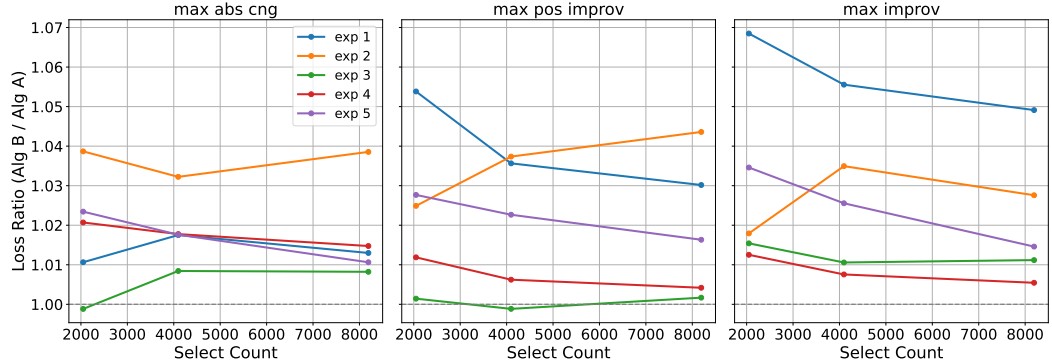

Figure 9: Ratio of test log-loss for Parallel ToV (Method B; Algorithm 2) relative to Interleaved ToV (Method A; Algorithm 1) for instruction tuning. Results use the SCORE+RANDOM selection strategy. Each subplot corresponds to one scoring strategy; lines denote different experimental setups in Table 1.

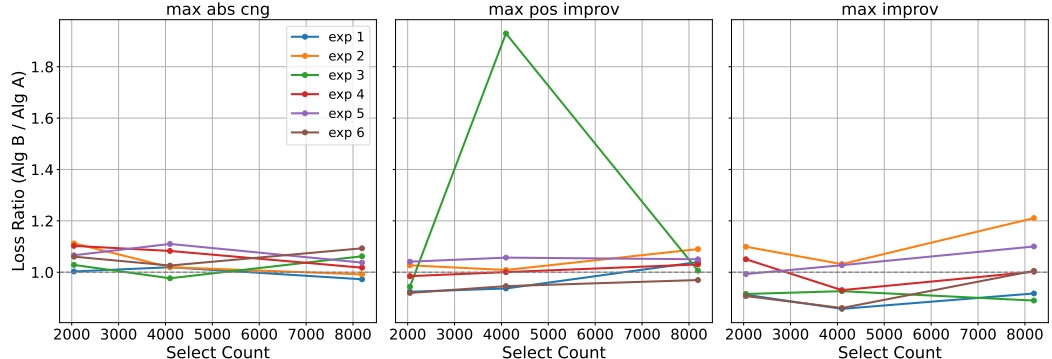

Figure 10: Ratio of test log-loss for Parallel ToV (Method B; Algorithm 2) relative to Interleaved ToV (Method A; Algorithm 1) for NER. Results use the SCORE+RANDOM selection strategy. Each subplot corresponds to one scoring strategy; lines denote different experimental setups in Table 2.

to a logistic model:

$$\Pr(y = 1 \mid x) = \frac{1}{1 + \exp(-x \cdot \theta)}, \quad \Pr(y = 0 \mid x) = 1 - \Pr(y = 1 \mid x).$$

We randomly sample a unit vector $\theta^*$ from the unit sphere to serve as the target direction. A second unit vector $\theta'$ is then drawn such that it lies at an angle $\gamma$ from $\theta^*$. In our experiments, we set $p = 10$ and $\gamma = \pi/2$.

The training pool consists of $N = 128 \times 1024$ samples, drawn independently from the mixture distribution:

$$\mathcal{D}_{\text{train}} = \frac{1}{2}\mathcal{P}_{\theta^*} + \frac{1}{2}\mathcal{P}_{\theta'}.$$

The validation and test sets contain $m_{\text{val}} = 1024$ and $m_{\text{tst}} = 10{,}000$ samples respectively, both drawn i.i.d. from the target distribution $\mathcal{P}_{\theta^*}$.

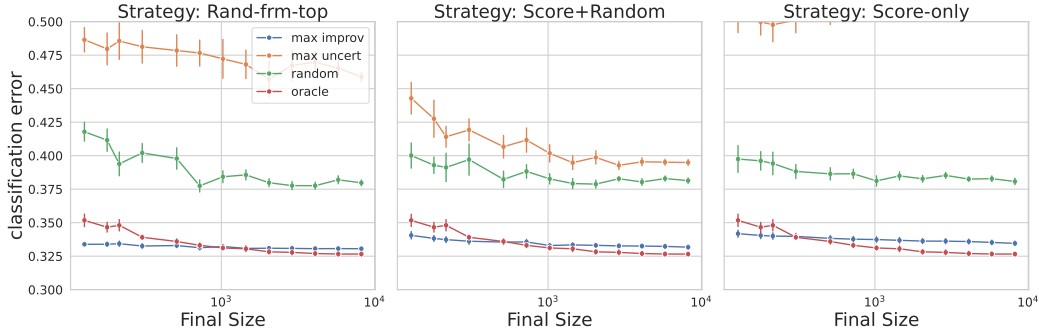

Figure 11: Data selection experiments with Parallel ToV for logistic regression on synthetic data in $d = 10$ dimensions (4 epochs of training). Each color corresponds to a distinct method to score data in the training pool, and each frame to a distinct method to use the score to form the selected set. Each symbol corresponds to the average of 10 experiments.

For scoring, we used Parallel ToV as described in Algorithm 2; similar experiments with Interleaved ToV produced comparable results, so we report only Parallel ToV here. Algorithm 2 does not specify the method to select data on the basis of scores. In Figure 11 we compare SCORE-ONLY and SCORE+RANDOM (already introduced above) with a third one RANDOM-FROM-TOP that selects at random from the top $50\%$ subset of data with highest scores.

The RANDOM-FROM-TOP method is included only for these synthetic logistic-regression experiments, for theoretical interest as by construction, the training pool contains half of its examples from the target distribution.

The base set used for initial training contains $|U| = 4 \times 1024$ examples.

For both the scoring model and the final model training, we used 4 epochs of batch gradient descent with a linear decay learning rate scheduler. The initial learning rate was set to 0.5. We used $\epsilon = \frac{1}{10}$ for adjusting the learning rate on validation examples.

The selected subset size $n$ was varied from 128 to 8192 in multiplicative steps of $\sqrt{2}$. All results are averaged over 10 independent runs. The final performance curves are presented in Figure 11.

## E  PROOF OF PROPOSITION 1

Throughout this proof, we denote by $C$ a generic constant that can depend on $c_0, C_1, M, C_\eta$ and whose value is allowed to change from line to line.

Letting $\boldsymbol{\Delta}_s(i) = \hat{\boldsymbol{\theta}}_s^{\text{bas}+i} - \hat{\boldsymbol{\theta}}_s^{\text{bas}}$, Eq. (11) yields

$$\begin{aligned}
\boldsymbol{\Delta}_{k+1}(i) &= \boldsymbol{\Delta}_k(i) - \eta m \nabla^2 \widehat{R}_U(\hat{\boldsymbol{\theta}}_k^{\text{bas}}) \boldsymbol{\Delta}_k(i) - \eta \nabla \ell(\hat{\boldsymbol{\theta}}_k^{\text{bas}}; \boldsymbol{x}_i) + \text{err}_k(i) \\
&= \boldsymbol{H}_k \boldsymbol{\Delta}_k(i) - \eta \nabla \ell(\hat{\boldsymbol{\theta}}_k^{\text{bas}}; \boldsymbol{x}_i) + \text{err}_k(i) \,,
\end{aligned} \tag{20}$$

where $\boldsymbol{H}_k$ is defined as in Eq. (14) and

$$\text{err}_k(i) := -\eta \big[ \nabla \ell(\hat{\boldsymbol{\theta}}_k^{\text{bas}+i}; \boldsymbol{x}_i) - \nabla \ell(\hat{\boldsymbol{\theta}}_k^{\text{bas}}; \boldsymbol{x}_i) \big] - \eta m \int_0^1 \big[ \nabla^2 \widehat{R}_U(\overline{\boldsymbol{\theta}}_k(z)) - \nabla^2 \widehat{R}_U(\hat{\boldsymbol{\theta}}_k^{\text{bas}}) \big] \boldsymbol{\Delta}_k(i) \mathrm{d}z \,,$$

where $\overline{\boldsymbol{\theta}}_k(z) = (1-z)\hat{\boldsymbol{\theta}}_k^{\text{bas}} + z\hat{\boldsymbol{\theta}}_k^{\text{bas}+i}$. By assumption $\boldsymbol{\theta} \mapsto \nabla \ell(\boldsymbol{\theta}; \boldsymbol{x}_i)$ and $\boldsymbol{\theta} \mapsto \nabla^2 \widehat{R}_U(\boldsymbol{\theta})$ are $M$-Lipschitz, whence

$$\begin{aligned}
\|\text{err}_k(i)\| &\leq \eta M \|\hat{\boldsymbol{\theta}}_k^{\text{bas}+i} - \hat{\boldsymbol{\theta}}_k^{\text{bas}}\| + \eta m M \|\hat{\boldsymbol{\theta}}_k^{\text{bas}+i} - \hat{\boldsymbol{\theta}}_k^{\text{bas}}\| \|\boldsymbol{\Delta}_k(i)\| \\
&= \eta M \|\boldsymbol{\Delta}_k(i)\| + \eta m M \|\boldsymbol{\Delta}_k(i)\|^2 \,.
\end{aligned} \tag{21}$$

Define $\boldsymbol{\Delta}_k^{\text{lin}}(i)$ by letting $\boldsymbol{\Delta}_k^{\text{lin}}(i) = \boldsymbol{0}$ and, for $k \geq 0$,

$$\boldsymbol{\Delta}_{k+1}^{\text{lin}}(i) = \boldsymbol{H}_k \boldsymbol{\Delta}_k^{\text{lin}}(i) - \eta \nabla \ell(\hat{\boldsymbol{\theta}}_k^{\text{bas}}; \boldsymbol{x}_i) \,. \tag{22}$$

Comparing with Eq. (20), we obtain

$$\big(\boldsymbol{\Delta}_{k+1}(i) - \boldsymbol{\Delta}_{k+1}^{\mathrm{lin}}(i)\big) = \boldsymbol{H}_k\big(\boldsymbol{\Delta}_{k+1}(i) - \boldsymbol{\Delta}_{k+1}^{\mathrm{lin}}(i)\big) + \mathrm{err}_k(\eta, m)\,,$$

$$\Rightarrow \ \boldsymbol{\Delta}_t(i) - \boldsymbol{\Delta}_t^{\mathrm{lin}}(i) = \sum_{s=0}^{t-1} \boldsymbol{M}_{t,s+1} \mathrm{err}_s(\eta, m)\,. \tag{23}$$

Since $\nabla^2 \widehat{R}_U(\hat{\boldsymbol{\theta}}_k^{\mathrm{bas}}) \succeq c_0 \boldsymbol{I}_d$, we have $\|\boldsymbol{H}_k\|_{\mathrm{op}} \le (1 - c_0 m \eta)$, and therefore

$$\|\boldsymbol{\Delta}_t(i) - \boldsymbol{\Delta}_t^{\mathrm{lin}}(i)\| \le \sum_{s=0}^{t-1} \|\boldsymbol{M}_{t,s+1}\|_{\mathrm{op}} \|\mathrm{err}_s(\eta, m)\|$$

$$\le \sum_{s=0}^{t-1} \big(1 - c_0 m \eta\big)^{t-s-1} \|\mathrm{err}_s(\eta, m)\|\,. \tag{24}$$

Further, from Eq. (22), and using $\|\nabla\ell(\hat{\boldsymbol{\theta}}_k^{\mathrm{bas}}; \boldsymbol{x}_i)\| \le C_1$, we get

$$\boldsymbol{\Delta}_t^{\mathrm{lin}}(i) = -\eta \sum_{s=0}^{t-1} \boldsymbol{M}_{t,s+1} \nabla\ell(\hat{\boldsymbol{\theta}}_s^{\mathrm{bas}}; \boldsymbol{x}_i)\,,$$

$$\Rightarrow \|\boldsymbol{\Delta}_t^{\mathrm{lin}}(i)\| \le C_1 \eta \sum_{s=0}^{t-1} \big(1 - c_0 m \eta\big)^{t-s-1} \le \frac{C}{m}\,. \tag{25}$$

Let $D_t(i) := \max_{s \le t} \|\boldsymbol{\Delta}_s(i)\|$, $E_t(i) := \max_{s \le t} \|\mathrm{err}_s(i)\|$. Using Eqs. (21), (24) and (25), we get

$$D_t(i) \le \frac{C}{m} + \frac{1}{c_0 m \eta} E_{t-1}(i)\,,$$

$$E_t(i) \le \eta M D_t(i) + \eta m M D_t(i)^2\,.$$

Using these inequalities together, we obtain, for all $m \ge m_0$ (and eventually adjusting the constant $C$)

$$D_t(i) \le \frac{C}{m}\,, \quad E_t(i) \le \frac{C\eta}{m}\,, \tag{26}$$

whence, using again Eq. (24), we get

$$\|\boldsymbol{\Delta}_t(i) - \boldsymbol{\Delta}_t^{\mathrm{lin}}(i)\| \le \frac{C}{m^2}\,. \tag{27}$$

Notice that we can rewrite

$$S_i^{\mathrm{lin}} = -\frac{1}{L} \sum_{s=1}^{L} \langle \nabla\widehat{R}_{\mathrm{val}}(\hat{\boldsymbol{\theta}}_s^{\mathrm{bas}}), \boldsymbol{\Delta}_s^{\mathrm{lin}}(i) \rangle\,, \tag{28}$$

whence, using the fact that $\|\nabla^2 \widehat{R}_{\mathrm{val}}(\overline{\boldsymbol{\theta}})\|_{\mathrm{op}} \le C$ for all $\overline{\boldsymbol{\theta}} \in [\hat{\boldsymbol{\theta}}_k^{\mathrm{bas}}, \hat{\boldsymbol{\theta}}_k^{\mathrm{bas}+i}]$ (this follows from the assumed bound $\|\nabla^2 \widehat{R}_{\mathrm{val}}(\hat{\boldsymbol{\theta}}_k^{\mathrm{bas}})\|_{\mathrm{op}} \le C_1$ and the Lipschitz property of $\boldsymbol{\theta} \mapsto \nabla^2 \widehat{R}_{\mathrm{val}}(\hat{\boldsymbol{\theta}})$), we get

$$\big|S_i - S_i^{\mathrm{lin}}\big| \le C \max_{s \le L} \|\boldsymbol{\Delta}_s(i)\|^2 + \frac{1}{L} \sum_{s=1}^{L} \big|\langle \nabla\widehat{R}_{\mathrm{val}}(\hat{\boldsymbol{\theta}}_s^{\mathrm{bas}}), \boldsymbol{\Delta}_s(i) - \boldsymbol{\Delta}_s^{\mathrm{lin}}(i) \rangle\big|$$

$$\le C \max_{s \le L} \|\boldsymbol{\Delta}_s(i)\|^2 + C \max_{s \le L} \|\boldsymbol{\Delta}_s(i) - \boldsymbol{\Delta}_s^{\mathrm{lin}}(i)\|$$

$$\le \frac{C}{m^2}\,,$$

and this completes the proof.

## F    PROOF OF THEOREM 1

Throughout this proof, we denote by $C$ a generic constant that can depend on $c_0, C_1, M, C_\eta$ and whose value is allowed to change from line to line.

### F.1 Bound on $\Upsilon_i$

The iteration for $\hat{\boldsymbol{\theta}}_k^{\mathsf{bas}}$ and $\hat{\boldsymbol{\theta}}_k^{\mathsf{bas},+}$, as specified by Algorithm 2, reads

$$\hat{\boldsymbol{\theta}}_{k+1}^{\mathsf{bas}} = \hat{\boldsymbol{\theta}}_k^{\mathsf{bas}} - \eta m \nabla \widehat{R}_U(\hat{\boldsymbol{\theta}}_k^{\mathsf{bas}}) \,, \tag{29}$$

$$\hat{\boldsymbol{\theta}}_{k+1}^{\mathsf{bas},+} = \hat{\boldsymbol{\theta}}_{0,k+1}^{\mathsf{bas},+} - \varepsilon \eta m_{\mathsf{val}} \nabla \widehat{R}_{\mathsf{val}}(\hat{\boldsymbol{\theta}}_{0,k+1}^{\mathsf{bas},+}) \,, \qquad \hat{\boldsymbol{\theta}}_{0,k+1}^{\mathsf{bas},+} = \hat{\boldsymbol{\theta}}_k^{\mathsf{bas},+} - \eta m \nabla \widehat{R}_U(\hat{\boldsymbol{\theta}}_k^{\mathsf{bas},+}) \,. \tag{30}$$

Letting $\boldsymbol{\Delta}_k := \hat{\boldsymbol{\theta}}_k^{\mathsf{bas},+} - \hat{\boldsymbol{\theta}}_k^{\mathsf{bas}}$, and $\boldsymbol{\Delta}_{0,k} := \hat{\boldsymbol{\theta}}_{0,k}^{\mathsf{bas},+} - \hat{\boldsymbol{\theta}}_k^{\mathsf{bas}}$, we obtain

$$\boldsymbol{\Delta}_{k+1} = \boldsymbol{\Delta}_{0,k+1} - \varepsilon \eta m_{\mathsf{val}} \nabla \widehat{R}_{\mathsf{val}}(\hat{\boldsymbol{\theta}}_{k+1}^{\mathsf{bas}}) + \mathsf{err}_{k+1}^{(1)} \,, \tag{31}$$

$$\boldsymbol{\Delta}_{0,k+1} = \boldsymbol{H}_k \boldsymbol{\Delta}_k + \mathsf{err}_k^{(2)} \,. \tag{32}$$

where, letting $\overline{\boldsymbol{\theta}}_{0,k+1}(z) = (1-z)\hat{\boldsymbol{\theta}}_{k+1}^{\mathsf{bas}} + z\hat{\boldsymbol{\theta}}_{0,k+1}^{\mathsf{bas},+}$ and $\overline{\boldsymbol{\theta}}_k(z) = (1-z)\hat{\boldsymbol{\theta}}_k^{\mathsf{bas}} + z\hat{\boldsymbol{\theta}}_k^{\mathsf{bas},+}$, we have

$$\mathsf{err}_{k+1}^{(1)} := -\eta \varepsilon m_{\mathsf{val}} \int_0^1 \nabla^2 \widehat{R}_{\mathsf{val}}(\overline{\boldsymbol{\theta}}_{0,k+1}(z)) \boldsymbol{\Delta}_{0,k+1} \, \mathrm{d}z \,,$$

$$\mathsf{err}_k^{(2)} := -\eta m \int_0^1 \big[ \nabla^2 \widehat{R}_U(\overline{\boldsymbol{\theta}}_k(z)) - \nabla^2 \widehat{R}_U(\hat{\boldsymbol{\theta}}_k^{\mathsf{bas}}) \big] \boldsymbol{\Delta}_k \, \mathrm{d}z \,.$$

Using the assumption that $\|\nabla^2 \widehat{R}_{\mathsf{val}}(\hat{\boldsymbol{\theta}}_{k+1}^{\mathsf{bas}}(z))\|_{\mathsf{op}} \le C$ and $\boldsymbol{\theta} \mapsto \nabla^2 \widehat{R}_{\mathsf{val}}(\boldsymbol{\theta})$ is $M$-Lipschitz, we get:

$$\|\mathsf{err}_{k+1}^{(1)}\| \le C\varepsilon \eta m_{\mathsf{val}} \big\{ \|\boldsymbol{\Delta}_{0,k+1}\| + \|\boldsymbol{\Delta}_{0,k+1}\|^2 \big\} \,. \tag{33}$$

On the other hand, since $\boldsymbol{\theta} \mapsto \nabla^2 \widehat{R}_U(\boldsymbol{\theta})$ is also $M$-Lipschitz, we have

$$\|\mathsf{err}_k^{(2)}\| \le \eta m M \|\boldsymbol{\Delta}_k\|^2 \,, \tag{34}$$

whence, using Eq. (32) and $\|\boldsymbol{H}_k\|_{\mathsf{op}} \le 1$

$$\|\boldsymbol{\Delta}_{0,k+1}\| \le \|\boldsymbol{\Delta}_k\| + \eta m M \|\boldsymbol{\Delta}_k\|^2$$
$$\Rightarrow \|\mathsf{err}_{k+1}^{(1)}\| \le C\varepsilon \eta m_{\mathsf{val}} \big\{ \|\boldsymbol{\Delta}_k\| + \|\boldsymbol{\Delta}_k\|^2 + \eta^2 m^2 \|\boldsymbol{\Delta}_k\|^4 \big\} \,, \tag{35}$$

where in the last line we used the assumption that $\eta m \le C_\eta$.

Substituting Eqs. (34) and (35) in Eq. (31), (32), we obtain (using again $\eta m \le C_\eta$)

$$\boldsymbol{\Delta}_{k+1} = \boldsymbol{H}_k \boldsymbol{\Delta}_k - \varepsilon \eta m_{\mathsf{val}} \nabla \widehat{R}_{\mathsf{val}}(\hat{\boldsymbol{\theta}}_{k+1}^{\mathsf{bas}}) + \mathsf{err}_k \,, \tag{36}$$

$$\|\mathsf{err}_k\| \le C\eta \varepsilon m_{\mathsf{val}} \big( \|\boldsymbol{\Delta}_k\| + \|\boldsymbol{\Delta}_k\|^4 \big) + C\eta m \|\boldsymbol{\Delta}_k\|^2 \,. \tag{37}$$

We define $\boldsymbol{\Delta}_k^{\mathsf{lin}} = \boldsymbol{0}$ and, for $k \ge 0$,

$$\boldsymbol{\Delta}_{k+1}^{\mathsf{lin}} = \boldsymbol{H}_k \boldsymbol{\Delta}_k^{\mathsf{lin}} - \varepsilon \eta m_{\mathsf{val}} \nabla \widehat{R}_{\mathsf{val}}(\hat{\boldsymbol{\theta}}_{k+1}^{\mathsf{bas}}) \,, \tag{38}$$

whence

$$\boldsymbol{\Delta}_t^{\mathsf{lin}} = -\varepsilon \eta m_{\mathsf{val}} \sum_{s=0}^{t-1} \boldsymbol{M}_{t,s+1} \nabla \widehat{R}_{\mathsf{val}}(\hat{\boldsymbol{\theta}}_{s+1}^{\mathsf{bas}}) \,, \qquad \boldsymbol{\Delta}_t - \boldsymbol{\Delta}_t^{\mathsf{lin}} = \sum_{s=0}^{t-1} \boldsymbol{M}_{t,s+1} \mathsf{err}_s \,. \tag{39}$$

Define $D_t := \max_{s \le t} \|\boldsymbol{\Delta}_s\|$, $E_t := \max_{s \le t} \|\mathsf{err}_s\|$. Using the fact that $\|\boldsymbol{M}_{t,s+1}\|_{\mathsf{op}} \le (1 - c_0 m \eta)^{t-s-1}$ and the assumption $\|\nabla \widehat{R}_{\mathsf{val}}(\hat{\boldsymbol{\theta}}_k^{\mathsf{bas}})\| \le C_1$, we get, from Eqs. (37), (39),

$$D_{t+1} \le \frac{C\varepsilon m_{\mathsf{val}}}{m} + \frac{C}{m\eta} E_t \,, \tag{40}$$

$$E_t \le C\eta \varepsilon m_{\mathsf{val}}(D_t + D_t^4) + C\eta m D_t^2 \,, \tag{41}$$

Using the assumption $\varepsilon m_{\mathsf{val}}/m \le c_*$, this is easily seen to imply

$$D_t \le C\frac{\varepsilon m_{\mathsf{val}}}{m} \,, \qquad E_t \le C\frac{(\varepsilon m_{\mathsf{val}})^2}{m}\eta \,. \tag{42}$$

Substituting in Eq. (39), we get

$$\|\mathbf{\Delta}_t - \mathbf{\Delta}_t^{\text{lin}}\| \le \sum_{s=0}^{t-1}(1 - c_0 m\eta)^{t-s-1}\|\text{err}_s\| \le C\Big(\frac{\varepsilon m_{\text{val}}}{m}\Big)^2. \tag{43}$$

The linearized score of Eq. (16) can be rewritten as

$$\Upsilon_i^{\text{lin}}\varepsilon = -\frac{1}{L}\sum_{s=1}^{L}\langle\nabla\ell(\hat{\boldsymbol{\theta}}_s^{\text{bas}}; \boldsymbol{x}_i), \mathbf{\Delta}_s^{\text{lin}}\rangle. \tag{44}$$

Using the fact that $\|\nabla\ell(\hat{\boldsymbol{\theta}}_k^{\text{bas}}; \boldsymbol{x}_i)\|, \|\nabla\ell(\hat{\boldsymbol{\theta}}_k^{\text{bas}}; \boldsymbol{x}_i)\|_{\text{op}} \le C_1$, we get

$$\big|\Upsilon_i(\varepsilon) - \Upsilon_i^{\text{lin}}\varepsilon\big| \le \frac{1}{L}\sum_{s=1}^{L}\Big|\ell(\hat{\boldsymbol{\theta}}_k^{\text{bas}}; \boldsymbol{x}_i) - \ell(\hat{\boldsymbol{\theta}}_k^{\text{bas},+}; \boldsymbol{x}_i) + \langle\nabla\ell(\hat{\boldsymbol{\theta}}_s^{\text{bas}}; \boldsymbol{x}_i), \mathbf{\Delta}_s^{\text{lin}}\rangle\Big| \tag{45}$$

$$\le \frac{C}{L}\sum_{s=1}^{L}\|\mathbf{\Delta}_s\|^2 + \frac{C}{L}\sum_{s=1}^{L}\|\mathbf{\Delta}_s - \mathbf{\Delta}_s^{\text{lin}}\| \tag{46}$$

$$\le C\Big(\frac{\varepsilon m_{\text{val}}}{m}\Big)^2, \tag{47}$$

### F.2 Bound on $\phi_i$

The iteration for $\hat{\boldsymbol{\theta}}_k^{\text{bas}}$ and $\hat{\boldsymbol{\theta}}_k^{\text{bas},+}$, as specified by Algorithm 2, reads

$$\hat{\boldsymbol{\theta}}_{k+1}^{\text{bas}} = \hat{\boldsymbol{\theta}}_k^{\text{bas}} - \eta m\nabla\widehat{R}_U(\hat{\boldsymbol{\theta}}_k^{\text{bas}}), \tag{48}$$

$$\hat{\boldsymbol{\theta}}_{k+1}^{\text{val}} = \hat{\boldsymbol{\theta}}_{k+1}^{\text{bas}} - \varepsilon\eta m_{\text{val}}\nabla\widehat{R}_{\text{val}}(\hat{\boldsymbol{\theta}}_{k+1}^{\text{bas}}). \tag{49}$$

Hence, we can rewrite

$$\phi_i^{\text{lin}}\varepsilon = -\frac{1}{L}\sum_{s=1}^{L}\langle\nabla\ell(\hat{\boldsymbol{\theta}}_s^{\text{bas}}; \boldsymbol{x}_i), \hat{\boldsymbol{\theta}}_s^{\text{val}} - \hat{\boldsymbol{\theta}}_s^{\text{bas}}\rangle.$$

Using the assumptions $\|\nabla^2\ell(\hat{\boldsymbol{\theta}}_s^{\text{bas}}; \boldsymbol{x}_i)\|_{\text{op}} \le C_1, \|\widehat{R}_{\text{val}}(\hat{\boldsymbol{\theta}}_k^{\text{bas}})\| \le C_1$, we obtain

$$\big|\phi_i(\varepsilon) - \phi_i^{\text{lin}}\varepsilon\big| \le \frac{1}{L}\sum_{s=1}^{L}\Big|\ell(\hat{\boldsymbol{\theta}}_s^{\text{val}}; \boldsymbol{x}_i) - \ell(\hat{\boldsymbol{\theta}}_s^{\text{bas}}; \boldsymbol{x}_i) - \langle\nabla\ell(\hat{\boldsymbol{\theta}}_s^{\text{bas}}; \boldsymbol{x}_i), \hat{\boldsymbol{\theta}}_s^{\text{val}} - \hat{\boldsymbol{\theta}}_s^{\text{bas}}\rangle\Big| \tag{50}$$

$$\le \frac{C}{L}\sum_{s=1}^{L}\|\hat{\boldsymbol{\theta}}_s^{\text{val}} - \hat{\boldsymbol{\theta}}_s^{\text{bas}}\|^2 \le C(\varepsilon\eta m_{\text{val}})^2. \tag{51}$$

The claim thus follows by recalling that $\eta \le C_\eta/m$.

## G Proof of Theorem 2

To lighten notation, we define $\boldsymbol{r}_k := \nabla\widehat{R}_{\text{val}}(\hat{\boldsymbol{\theta}}_k^{\text{bas}})$ and $\boldsymbol{v}_k(i) := \nabla\ell(\hat{\boldsymbol{\theta}}_k^{\text{bas}}; \boldsymbol{x}_i)$.

For any $L, L_1 \in \mathbb{Z}$, we have

$$\Upsilon_i^{\text{lin}}(L) = \Upsilon_i^{\text{lin},0}(L) + \Upsilon_i^{\text{lin},1}(L) + \Upsilon_i^{\text{lin},2}(L) + \Upsilon_i^{\text{lin},3}(L),$$

$$\Upsilon_i^{\text{lin},0}(L) := \frac{\eta m_{\text{val}}}{L} \sum_{0 \le t < s \le L_0} \langle \boldsymbol{r}_{t+1}, \boldsymbol{M}_{s,t+1}^{\mathsf{T}} \boldsymbol{v}_s(i) \rangle,$$

$$\Upsilon_i^{\text{lin},1}(L) := \frac{\eta m_{\text{val}}}{L} \sum_{0 \le t \le L_0, L_0 < s \le L} \langle \boldsymbol{r}_{t+1}, \boldsymbol{M}_{s,t+1}^{\mathsf{T}} \boldsymbol{v}_s(i) \rangle,$$

$$\Upsilon_i^{\text{lin},2}(L) := \frac{\eta m_{\text{val}}}{L} \sum_{L_0 < t < s \le L_0 : |s-t| \ge L_1} \langle \boldsymbol{r}_{t+1}, \boldsymbol{M}_{s,t+1}^{\mathsf{T}} \boldsymbol{v}_s(i) \rangle,$$

$$\Upsilon_i^{\text{lin},2}(L) := \frac{\eta m_{\text{val}}}{L} \sum_{L_0 < t < s \le L_0 : |s-t| \ge L_1} \langle \boldsymbol{r}_{t+1}, \boldsymbol{M}_{s,t+1}^{\mathsf{T}} \boldsymbol{v}_s(i) \rangle,$$

$$\Upsilon_i^{\text{lin},3}(L) := \frac{\eta m_{\text{val}}}{L} \sum_{L_0 < t < s \le L_0 : |s-t| < L_1} \langle \boldsymbol{r}_{t+1}, \boldsymbol{M}_{s,t+1}^{\mathsf{T}} \boldsymbol{v}_s(i) \rangle.$$

Since by continuity we have $\lim_{k \to \infty} \nabla^2 \widehat{R}_U(\hat{\boldsymbol{\theta}}_k^{\text{bas}}) = \boldsymbol{Q}_\infty$, for any $\delta \in (0, 1/2)$, we can choose $L_0$ large enough so that $(1 - \delta)\boldsymbol{Q}_\infty \preceq \nabla^2 \widehat{R}_U(\hat{\boldsymbol{\theta}}_k^{\text{bas}}) \preceq (1 + \delta)\boldsymbol{Q}_\infty$ for all $k > L_0$. In particular there exists $c_0 > 0$ (independent of $\varepsilon$) such that $\|\boldsymbol{H}_k\|_{op} \le (1 - c_0 m\eta)$ for all $k > L_0$.

Clearly $|\Upsilon_i^{\text{lin},0}(L)| \le C(L_0)/L \to 0$ as $L \to \infty$. Further

$$\left| \Upsilon_i^{\text{lin},1}(L) \right| \le \frac{C\eta m_{\text{val}}}{L} \sum_{0 \le t \le L_0, L_0 < s \le L} \|\boldsymbol{M}_{s,t+1}\|_{op}$$

$$\le \frac{C\eta m_{\text{val}}}{L} \sum_{0 \le t \le L_0, L_0 < s \le L} (1 - c_0 m\eta)^{s-t-1}$$

$$\le \frac{C\eta m_{\text{val}}}{L} \frac{L_0}{c_0 m\eta} \to 0.$$

Finally, by increasing $L_0$, we can ensure that, for $k > L_0$, $\|\boldsymbol{H}_k - \boldsymbol{H}_\infty\|_{op} \le \delta$, $\|\boldsymbol{r}_k - \boldsymbol{r}_\infty\| \le \delta$, $\|\boldsymbol{v}_k(i) - \boldsymbol{v}_\infty(i)\| \le \delta$ (where $\boldsymbol{H}_\infty = \boldsymbol{I} - \eta m \boldsymbol{Q}_\infty$ and $\boldsymbol{r}_\infty, \boldsymbol{v}_\infty(i)$). Hence

$$\left| \langle \boldsymbol{r}_{t+1}, \boldsymbol{M}_{s,t+1}^{\mathsf{T}} \boldsymbol{v}_s(i) \rangle - \langle \boldsymbol{r}_\infty, \boldsymbol{H}_\infty^{s-t-1} \boldsymbol{v}_\infty(i) \rangle \right| \le C|t - s + 1|(1 - c_0 m\eta)^{s-t-1} \delta.$$

Therefore, letting

$$\tilde{\Upsilon}_i^{\text{lin},2}(L) := \frac{\eta m_{\text{val}}}{L} \sum_{L_0 < t < s \le L} \langle \boldsymbol{r}_\infty, \boldsymbol{H}_\infty^{s-t-1} \boldsymbol{v}_\infty(i) \rangle, \tag{52}$$

we have

$$\left| \Upsilon_i^{\text{lin},2}(L) - \tilde{\Upsilon}_i^{\text{lin},2}(L) \right| \le \frac{\eta m_{\text{val}}}{L} \sum_{L_0 < t < s \le L} \left| \langle \boldsymbol{r}_{t+1}, \boldsymbol{M}_{s,t+1}^{\mathsf{T}} \boldsymbol{v}_s(i) \rangle - \langle \boldsymbol{r}_\infty, \boldsymbol{H}_\infty^{s-t-1} \boldsymbol{v}_\infty(i) \rangle \right|$$

$$\le \frac{\eta m_{\text{val}}}{L} \sum_{L_0 < t < s \le L} C|t - s + 1|(1 - c_0 m\eta)^{s-t-1} \delta$$

$$\le \eta m_{\text{val}} \cdot \frac{1}{(c_0 m\eta)^2} \delta.$$

Finally, using again $|\langle \boldsymbol{r}_\infty, \boldsymbol{H}_\infty^{s-t-1} \boldsymbol{v}_\infty(i) \rangle| \le (1 - c_0 m\eta)^{s-t-1}$, we have

$$\lim_{L \to \infty} \tilde{\Upsilon}_i^{\text{lin},2}(L) = \lim_{L \to \infty} \frac{\eta m_{\text{val}}}{L} \sum_{L_0 < t \le L} \sum_{s=t+1}^{\infty} \langle \boldsymbol{r}_\infty, \boldsymbol{H}_\infty^{s-t-1} \boldsymbol{v}_\infty(i) \rangle$$

$$= \eta m_{\text{val}} \sum_{k=0}^{\infty} \langle \boldsymbol{r}_\infty, \boldsymbol{H}_\infty^k \boldsymbol{v}_\infty(i) \rangle$$

$$= \eta m_{\text{val}} \langle \boldsymbol{r}_\infty, (\boldsymbol{I} - \boldsymbol{H}_\infty)^{-1} \boldsymbol{v}_\infty(i) \rangle$$

$$= \frac{m_{\text{val}}}{m} \langle \boldsymbol{r}_\infty, \boldsymbol{Q}_\infty^{-1} \boldsymbol{v}_\infty(i) \rangle.$$

This finishes the proof of the part of Eq. (18) which concerns the limit of $\Upsilon_i^{\text{lin}}$. The calculation of $\lim_{L\to\infty} S_i^{\text{lin}}(L)$ is completely analogous and we omit it.

## H  PROOF OF THEOREM 3

To simplify notations, we write $y_j = y(\boldsymbol{x}_j)$ for the response variables and $\boldsymbol{\psi}_j = \boldsymbol{\psi}(\boldsymbol{x}_j)$ for the feature vectors. Similarly, for the $y_j^{\text{val}} = y(\boldsymbol{z}_j^{\text{val}})$, $\boldsymbol{\psi}(\boldsymbol{z}_j^{\text{val}}) = \boldsymbol{\psi}_j^{\text{val}}$.

With these notations, we have $\boldsymbol{H}_k = \boldsymbol{H}$ independent of $k$ and

$$\nabla \ell(\hat{\boldsymbol{\theta}}; \boldsymbol{x}_i) = -(y_i - \langle \boldsymbol{\psi}_i, \boldsymbol{\theta} \rangle) \boldsymbol{\psi}_i \,, \tag{53}$$

$$\nabla \widehat{R}_{\text{val}}(\boldsymbol{\theta}) = -\frac{1}{m} \boldsymbol{\Psi}^{\mathsf{T}} (\boldsymbol{y} - \boldsymbol{\Psi}\boldsymbol{\theta}) \,, \tag{54}$$

$$\boldsymbol{H} = \boldsymbol{I} - \eta \boldsymbol{\Psi}^{\mathsf{T}} \boldsymbol{\Psi} \,. \tag{55}$$

Hence,

$$\Upsilon_i^{\text{lin}} = \frac{\eta m_{\text{val}}}{L} \sum_{0 \le t < s \le L} r_s(i) \langle \boldsymbol{\Psi} \boldsymbol{r}_{t+1}^{\text{val}}, \boldsymbol{H}^{s-t-1} \boldsymbol{\psi}_i \rangle \,, \tag{56}$$

$$\boldsymbol{r}_t^{\text{val}} := \boldsymbol{y}^{\text{val}} - \boldsymbol{\psi}^{\text{val}} \hat{\boldsymbol{\theta}}_{t+1}^{\text{bas}} \,, \qquad r_s(i) := y_i - \langle \boldsymbol{\psi}_i, \hat{\boldsymbol{\theta}}_s^{\text{bas}} \rangle \tag{57}$$

Since $\boldsymbol{P}_{\boldsymbol{\Psi}}$ is the projector onto the null-space of $\boldsymbol{H}$, and by our choice of $\eta$, we have $\boldsymbol{H} = \boldsymbol{P}_{\boldsymbol{\Psi}} + \boldsymbol{H}_\perp$, where the row/column space of $\boldsymbol{H}_\perp$ is orthogonal to the one of $\boldsymbol{P}_{\boldsymbol{\Psi}}$ and $\|\boldsymbol{H}_\perp\|_{\text{op}} = (1 - c_\psi \eta) \in [0, 1)$. As a consequence $\|\boldsymbol{H}^{s-t-1} - \boldsymbol{P}_{\boldsymbol{\Psi}}\|_{\text{op}} \le (1 - c_\psi \eta)^{s-t-1}$.

Define

$$\tilde{\Upsilon}_i^{\text{lin}} := \frac{\eta m_{\text{val}}}{L} \sum_{0 \le t < s \le L} r_s(i) \langle \boldsymbol{\Psi} \boldsymbol{r}_{t+1}^{\text{val}}, \boldsymbol{P}_{\boldsymbol{\Psi}} \boldsymbol{\psi}_i \rangle \,. \tag{58}$$

Then we have

$$\left| \frac{1}{L} \Upsilon_i^{\text{lin}} - \frac{1}{L} \tilde{\Upsilon}_i^{\text{lin}} \right| \le \frac{\eta m_{\text{val}}}{L^2} \sum_{0 \le t < s \le L} \left| r_s(i) \langle \boldsymbol{\Psi} \boldsymbol{r}_{t+1}^{\text{val}}, \boldsymbol{P}_{\boldsymbol{\Psi}} \boldsymbol{\psi}_i \rangle \right|$$

$$\le \frac{\eta m_{\text{val}}}{L^2} \sum_{0 \le t < s \le L} |r_s(i)| \, \|\boldsymbol{\Psi} \boldsymbol{r}_{t+1}^{\text{val}}\| \, \|\boldsymbol{H}^{s-t-1} - \boldsymbol{P}_{\boldsymbol{\Psi}}\|_{\text{op}} \, |\boldsymbol{\psi}_i|$$

$$\overset{(a)}{\le} C \frac{\eta m_{\text{val}}}{L^2} \sum_{0 \le t < s \le L} (1 - c_\psi \eta)^{s-t-1}$$

$$\le C \frac{\eta m_{\text{val}}}{L} \frac{1}{c_\psi \eta} \overset{L\to\infty}{\longrightarrow} 0 \,.$$

In $(a)$, we used the fact that $\lim_{t\to\infty} \hat{\boldsymbol{\theta}}_t^{\text{bas}} = \hat{\boldsymbol{\theta}}$ (Bartlett et al., 2021), and therefore $|r_s(i)|$, $\|\boldsymbol{\Psi} \boldsymbol{r}_{t+1}^{\text{val}}\|$ remain bounded as $s, t \to \infty$.

In view of the above, $\lim_{L\to\infty} \Upsilon_i^{\text{lin}}/L = \lim_{L\to\infty} \tilde{\Upsilon}_i^{\text{lin}}/L$. For the latter, we have

$$\lim_{L\to\infty} \frac{1}{L} \tilde{\Upsilon}_i^{\text{lin}} = \lim_{L\to\infty} \frac{\eta m_{\text{val}}}{L^2} \sum_{0 \le t < s \le L} r_s(i) \langle \boldsymbol{\Psi} \boldsymbol{r}_{t+1}^{\text{val}}, \boldsymbol{P}_{\boldsymbol{\Psi}} \boldsymbol{\psi}_i \rangle$$

$$= \lim_{L\to\infty} \frac{\eta m_{\text{val}}}{L^2} \sum_{L_0 \le t < s \le L} r_s(i) \langle \boldsymbol{\Psi} \boldsymbol{r}_{t+1}^{\text{val}}, \boldsymbol{P}_{\boldsymbol{\Psi}} \boldsymbol{\psi}_i \rangle$$

$$= \lim_{L\to\infty} \frac{\eta m_{\text{val}}}{L^2} \sum_{L_0 \le t < s \le L} r(i) \langle \boldsymbol{\Psi} \boldsymbol{r}^{\text{val}}, \boldsymbol{P}_{\boldsymbol{\Psi}} \boldsymbol{\psi}_i \rangle + \text{err}(L_0, L) \,,$$

where

$$|\text{err}(L_0, L)| \le C \sup_{s \ge L_0} \|r_s(i) - r(i)\| + C \sup_{t \ge L_0} \|\boldsymbol{r}_t^{\text{val}} - \boldsymbol{r}_{t+1}^{\text{val}}\| \,. \tag{59}$$

Since $\lim_{t\to\infty} \hat{\boldsymbol{\theta}}_t^{\mathrm{bas}} = \hat{\boldsymbol{\theta}}$, we have $\lim_{L_0\to\infty} \limsup_{L\to\infty} \mathrm{err}(L_0, L) = 0$. Therefore,

$$\lim_{L\to\infty} \frac{1}{L}\tilde{\Upsilon}_i^{\mathrm{lin}} = \lim_{L_0\to\infty} \lim_{L\to\infty} \frac{\eta m_{\mathrm{val}}}{L^2} \sum_{L_0 \le t < s \le L} r(i)\langle \boldsymbol{\Psi r}^{\mathrm{val}}, \boldsymbol{P_\Psi \psi}_i \rangle$$

$$= \frac{1}{2}\eta m_{\mathrm{val}} r(i)\langle \boldsymbol{\Psi r}^{\mathrm{val}}, \boldsymbol{P_\Psi \psi}_i \rangle.$$

This proves the limit for $\Upsilon_i^{\mathrm{lin}}(L)$ in Eq. (19).

The limit of $S_i^{\mathrm{lin}}(L)$ is computed essentially by the same argument and we omit the derivation.

## I  GRADIENT ACCUMULATION IN ToV

A potential concern in Train-on-Validation (ToV) scoring is that repeated updates on the validation set across scoring checkpoints could accumulate on the same model parameters and lead to overfitting of the selection policy to $\boldsymbol{Z}^{\mathrm{val}}$. In Interleaved ToV, however, each scoring model is initialized from a base-model checkpoint that has not previously been updated on the validation set and is then updated on $\boldsymbol{Z}^{\mathrm{val}}$ only once. Thus, validation gradients do not accumulate on the same parameters across epochs. In Parallel ToV, validation-updated models continue training and may, in principle, accumulate such effects; in practice, this is mitigated by using a much smaller learning rate for validation training. Nevertheless, Parallel ToV may remain more susceptible to validation overfitting, which could contribute to its slightly inferior performance relative to Interleaved ToV.

## J  LIMITATION

The core idea of "train-on-validation" impacting training examples is general, but the specific scoring function $F(.)$ and aggregation strategy might need adaptation for different problem settings.

The SCORE+RANDOM selection strategy often outperformed SCORE-ONLY in our experiments, suggesting that diversity plays an important role beyond simply selecting the "most affected" examples. While this is a practical improvement, it also indicates that our current scoring mechanism might not fully capture the optimal diversity or coverage needed for effective generalization. It will be interesting to explore more sophisticated diversity-aware scoring or selection mechanisms that explicitly balance our scoring methods with representation across the data space.

Although we mitigated bias toward shorter examples through length-based binning, a more refined length-normalization or task-specific weighting might further enhance the selection process. Furthermore, it will be interesting to see if the performance of our strategies improves further compared to random selection when the learning rate is also tuned for these strategies, rather than just for random selection.

Finally, our theoretical analysis relies on stylized settings that are plausible for simple models but may not hold in many large-scale applications.

## K  MODELS AND DATASETS INFORMATION

### K.1  DATASET INFORMATION

- Slim Orca:
    - Link
    - Citations-Longpre et al. (2023); Mukherjee et al. (2023); Lian et al. (2023)
    - Licence: mit
- Alpaca GPT-4:
    - Paper:Peng et al. (2023)
    - Repository
    - Link

- – Licence: cc-by-nc-4.0
- Alpaca GPT-3.5:
  - – Paper: Taori et al. (2023)
  - – Link
  - – Licence: cc-by-nc-4.0
- Multinerd:
  - – Paper: Tedeschi & Navigli (2022)
  - – Link
  - – Licence: cc-by-nc-sa-4.0
- Ai4p:
  - – Link
  - – Licence: link
- C4 dataset:
  - – Link
  - – Labeled for NER task using llms.
  - – Licence: terms of use
- Syn-Big:
  - – Synthetically generated by us using llms.
  - – Proprietary dataset

## K.2 PRETRAINED MODEL INFORMATION

- Meta-Llama-3-8B AI@Meta (2024)
  - – Link
  - – License: llama3
- xlm-roberta-base Conneau et al. (2019)
  - – Link
  - – License: mit

## K.3 COMPUTE RESOURCES INFORMATION

All experiments were conducted on a single machine equipped with 8 NVIDIA A100 (80 GB) GPUs and 128 AMD EPYC 9354 (32-core) CPUs.

Instruction-tuning experiments across all five setups, with 10 runs per setup and including hyper-parameter search, required approximately two weeks of wall-clock time using parallel execution across GPUs. NER experiments completed within 4–5 days under the same setup. Logistic regression experiments were lightweight and required less than 30 minutes in total.

