# OpenReview forum: "Train on Validation (ToV): Fast data selection with applications to fine-tuning"
_ICLR.cc/2026/Conference — ICLR 2026 Poster_

### Official Review · Reviewer_YYKn · 2025-10-25

**Soundness:** 2
**Presentation:** 3
**Contribution:** 3
**Rating:** 6
**Confidence:** 4

**Summary:**

The paper proposes a method for selecting task-specific fine-tuning data using a validation set representative of the target distribution. The approach finetunes a model on the validation set and performs inference on candidate training samples, selecting those whose losses change the most. Experiments demonstrate that the method outperforms uncertainty-based selection and a recent task-specific selection method. Finally, the authors further provide a theoretical justification for its effectiveness.

**Strengths:**

- Training on the validation set introduces a new perspective in task-specific data selection.
- The method is conceptually simple and readily implementable.
- A theoretical analysis is presented, providing provable error bounds.
- The manuscript is well structured and written with clarity.

**Weaknesses:**

- The paper claims that the proposed method avoids the computational burden of computing influence functions, but does not provide any empirical or theoretical evidence. It is unclear how expensive the proposed method is compared to training on the full set or influence-based selection.
- Only log loss is reported in the evaluation. For instruction-following and NER tasks, log loss does not adequately capture output quality. Metrics such as F1 (for NER) or semantic similarity and human preference scores (for instruction-following) would provide a more meaningful assessment.
- A gap exists between the empirically evaluated method and the theoretical formulation, and the paper does not make the connection between them explicit.

**Questions:**

- What is the purpose of fine-tuning on a small random subset before fine-tuning on the validation set? Additionally, why is it necessary to repeat this process for multiple iterations?
- In exp 3 and NER tasks, why does LESS perform so badly as a task-specific selection method?

---

> ### Author Response · Authors · 2025-11-21
>
> Thank you for your feedback on the paper and for highlighting the conceptual simplicity of the method, our theoretical contribution, the clarity and structure of the manuscript, and the new perspective that training on the validation set brings to data selection. We respond to each point below.
>
> 1. We noted in the introduction (lines 99–101) that computing per-example influence directly is prohibitively expensive. Both our approach and LESS provide efficient approximations to influence-style scores and are far cheaper than exact influence computation. Among these two approximation methods, we found ToV to be substantially faster and more memory-efficient across both the instruction-tuning and NER setups. Using the official LESS implementation released by the authors, we measured runtime and memory usage under identical hardware (mean over 5 runs). The results are summarized below:
> | Setting             | Method | Runtime        | Memory Usage                                      |
> |---------------------|--------|----------------|---------------------------------------------------|
> | Instruction Tuning  | LESS   | ~4 hours       | ~4.9 GB (0.9 GB checkpoints + ~4 GB gradients)    |
> | Instruction Tuning  | ToV    | ~2 hours       | ~1.84 GB total (0.141 GB scores)                  |
> | NER                 | LESS   | > 45 minutes   | ~4 GB (gradients dominate)                        |
> | NER                 | ToV    | ~8 minutes     | < 0.25 GB total                                   |
>
> All measurements correspond to Setup 3 for both IT and NER; the results for the remaining setups show the same trend. We will include the full table in the final version of the paper.
>
> Finally we imagine that by “training on the full set”, the referee means training without data selection. In this paper we focus on improving performances at fixed size of the selected set. We point out that
> (i) The complexity of our data selection method scales  like  no. of epoch of scoring model x inference time per sample x N_train. In contrast, training on the whole set scales like number of epochs for final training x gradient computation per samples x N_train. Hence the latter can be orders of magnitude more expensive.
> (ii) In industry scale applications, data selection is common practice for a number of reasons, eg instance data storage constraints.
>
> 2. Both our data selection method  and previous ones are explicitly designed to minimize cross-entropy on the validation set. As a consequence, log-loss is the correct metric for evaluating the quality of the different methods. While eventually we might be interested in downstream performance, this can be affected by other factors that are not directly related to the quality of data selection by itself.
>  LESS  observes that reductions in cross-entropy typically coincide with improvements in accuracy (see point 3 in their limitations section). For the NER task we also computed tokenwise F1 scores and observed similar consistent improvements as log loss; following your suggestion we will include these results in the final version. For the instruction-tuning task, there is no single uniform accuracy-style metric across our train and test datasets, so we reported log-loss/perplexity, which is the standard evaluation metric for language models. We leave evaluating the improvement in various downstream tasks for future work.
>
> 3. We emphasize that results for Method B are included in the appendix. In our experiments, a simplified version of Method B (Method A) consistently worked better in practice. The two methods are closely related and in fact coincide when the number of epochs is one. Method A also avoids repeated updates on the validation set, which helps reduce overfitting in practice.
>
> Answers:
> 1. The purpose of briefly fine-tuning on a random subset before the validation update is to evaluate the advantage of adding a training example at different stages of training. We choose a random set of examples and perform the same total number of steps as in the final training on the selected data to ensure that the scoring reflects how useful each example would be throughout training.
>
> 2. The underlying reason for the poor performance in the settings you mentioned remains unclear to us.

---

> > ### Comment · Reviewer_YYKn · 2025-11-21
> >
> > Thank you for the clarification. Could you additionally provide downstream NER performance numbers? Also, I’d appreciate more detail on the simplifications you applied to Method A vs Method B, and perhaps consider giving the methods more informative names.

---

> > > ### Author Response · Authors · 2025-11-22
> > >
> > > Thank you for the follow-up questions. We address each point below.
> > >
> > > 1. F1 plots
> > >
> > >    We have updated the supplementary material to include token-level F1 plots for all NER setups. These results show that our method improves F1 scores compared to random selection in all six settings. We did not originally compute F1 for LESS, but we will rerun the experiments and include those numbers in the final version. Given LESS’s weaker performance on this task relative to random selection, we do not expect it to provide a competitive F1 baseline for these setups, but we will report the results for completeness.
> > >
> > > 2. Method A vs Method B.
> > >
> > >    The key distinction lies in how the validation update interacts with the training trajectory for the scoring model, and in the computational cost of each cycle.
> > >
> > >    a. Method A
> > >
> > >        i. Performs a single baseline training run on the base subset U.
> > >
> > >        ii. At each epoch, we branch from this trajectory, take a one-epoch validation update only to compute scores, and then discard the branch.
> > >
> > >        iii. The validation updates never accumulate.
> > >
> > >        iv. This makes Method A more efficient and empirically more stable for LLM fine-tuning.
> > >
> > >    b. Method B
> > >
> > >        i. Maintains two trajectories per cycle:
> > >
> > >            • A baseline trajectory that trains only on U, and
> > >
> > >            • A second trajectory that trains on U and then on the validation set.
> > >
> > >        ii. Validation updates accumulate across cycles on the second trajectory.
> > >
> > >        iii. For the first epoch of the scoring model it is identical to Method A.
> > >
> > >        iv. This design matches the theoretical recursion analyzed in Section 3 and also improves over random selection, but it is generally less effective empirically than Method A.
> > >
> > > 3. Naming.
> > >
> > >    We thank the reviewer for suggesting the use of more informative names for these methods. We agree and will rename:
> > >
> > >        a. Method A → Interleaved ToV
> > >
> > >        b. Method B → Parallel ToV
> > >    in the final version.

---

### Official Review · Reviewer_5rLJ · 2025-10-31

**Soundness:** 3
**Presentation:** 3
**Contribution:** 3
**Rating:** 6
**Confidence:** 5

**Summary:**

This paper studies how to efficiently select training samples so that the resulting model performs better on a small validation set drawn from a target distribution. The authors observe that standard training-sample scoring methods rely on per-sample gradient computations over both the training and validation sets. To reduce this cost, they propose an approximation based on the change in training samples' loss after performing a gradient update using the full validation set. Two algorithms are developed based on this insight—the second being theoretically guaranteed but empirically less effective. Experimental results demonstrate consistent performance improvements over two baseline methods.

**Strengths:**

1. The motivation of this paper is strong, especially for scenarios where some existing packages do not support per-sample gradient computation.
2. The observation of train–validation symmetry is interesting. It is also a good choice to put this point in the introduction section, as it helps readers quickly grasp the core insight of the paper.
3. The proposed method could be applied to certain LLM-related tasks, such as instruction tuning and named entity recognition.
4. The theoretical analyses and experiments look good and convincing.

**Weaknesses:**

1. Although the train–validation symmetry is an interesting observation, it is established under the assumption of independence of x. In Proposition 1, the Hessian matrix is expected to be of order 1, which supports the utility of the proposed estimator. However, it would be beneficial to discuss this point in more detail, particularly in relation to influence functions and data Shapley. I understand that Strategy 2, which increases diversity via random sampling, can heuristically mitigate this issue to some extent.
2. The proposed method can also be interpreted as a task-dependent data pruning technique. From this perspective, it would be helpful to include a discussion comparing it with representative works in this area.
3. The original problem can be formulated as a bi-level optimization problem, typically with continuous sample weights. I did not look into whether Xia et al. (2024) is following this idea, but the current baseline choices seem somewhat weak in comparison.
4. It would be better to experimentally quantify the efficiency (how fast) of the proposed method, as this is claimed as the key contribution.

**Questions:**

1. Instead of selecting important samples, why not consider applying target-aware domain generalization techniques here, especially in the context of LLM tuning?
2. Since both training and validation sets are accessible, how about directly measuring sample similarity between them to guide selection?

---

> ### Author Response · Authors · 2025-11-21
>
> Thank you for your careful review and for highlighting the strength of our motivation, the clarity of the train–validation symmetry, and the usefulness of our method for LLM fine-tuning tasks such as instruction tuning and NER. We address each of your comments below.
> 1. First note that no independence assumption (indeed no probabilistic assumption) is made in the derivation of the train-validation duality, which instead has uniquely deterministic assumptions. We obtain three type of results: (i) First order Taylor approximation of the output of our algorithm; (ii) General relation between influence functions and output of our algorithm (in terms of the matrix M); (iii) Asymptotically exact train-validation symmetry. As the reviewer points out, we assume bounds on the Hessian of the empirical risk, but similar conditions are implicitly assumed by all influence-based methods (eg LESS or TRAK). Those papers do not state explicitly such assumptions simply because they do not prove quantitative bounds.
> We agree that the assumptions (in particular on the Hessian of the empirical risk) for theoretical derivations are stronger than what one would hope, and related to assumptions that justify the use of influence functions. We will make sure to emphasize limitations and connections.
> At the same time, we show in Section 3 that for some non-trivial cases, with explicit assumptions, our method can be justified rigorously.
>
> 2. Our approach is different from standard task-dependent data pruning. In our setting, there is no single well-defined supervised task; instruction-tuning datasets consist of prompts drawn from multiple distributions. For example, Alpaca-3.5 and Alpaca-4 include similar instructions, but the latter provides higher-quality responses. Likewise, in the NER experiments, all datasets correspond to the same underlying token-classification objective, making conventional pruning heuristics less applicable. Following your suggestion, we will add a short discussion comparing our approach with existing pruning methods.
>
> 3. Data selection can indeed be formulated as a bi-level optimization problem. A naive implementation, however, is often prohibitively expensive. Recent DataModel-style approaches (e.g., [1]) have made such formulations more tractable, but they remain considerably more complex, have no publicly available code for dataset-level selection, and are known to be sensitive to hyperparameters, making a reliable comparison difficult. In addition, these methods are still substantially more computationally expensive than both LESS and our approach.
> 	[1] Optimizing ML Training with Metagradient Descent.
>
> 4. We found ToV to be substantially faster and more memory-efficient across both the instruction-tuning and NER setups. Using the official LESS implementation released by the authors, we measured runtime and memory usage under identical hardware (mean over 5 runs). The results are summarized below:
> | Setting             | Method | Runtime        | Memory Usage                                      |
> |---------------------|--------|----------------|---------------------------------------------------|
> | Instruction Tuning  | LESS   | ~4 hours       | ~4.9 GB (0.9 GB checkpoints + ~4 GB gradients)    |
> | Instruction Tuning  | ToV    | ~2 hours       | ~1.84 GB total (0.141 GB scores)                  |
> | NER                 | LESS   | > 45 minutes   | ~4 GB (gradients dominate)                        |
> | NER                 | ToV    | ~8 minutes     | < 0.25 GB total                                   |
>
> All measurements correspond to Setup 3 for both IT and NER; the results for the remaining setups show the same trend. We will include the full table in the final version of the paper.
>  We also note that both ToV and LESS are far cheaper than directly estimating per-example influence (as discussed in the introduction), while ToV achieves consistently stronger empirical performance.
>
> Answers
> 1. As mentioned above, target-aware domain generalization techniques will not apply for the settings considered in our experiments.
> 2. In our setting similarity between the examples is less obvious, because of the above mentioned reasons. LESS calculated embedding based-similarity to select as baselines and found them to perform poorly, often worse than random selection.

---

### Official Review · Reviewer_E8Vn · 2025-10-31

**Soundness:** 3
**Presentation:** 2
**Contribution:** 3
**Rating:** 6
**Confidence:** 3

**Summary:**

Train on Validation (ToV) is a simple data-selection method for fine-tuning. Leveraging from a principle they drove in the paper, train–validation symmetry, they built a fast scoring rule that does not depend on the per-example gradients, or Hessian-vector product, as opposed to prior works. They simply compute each pool example’s loss before and after a brief train-on-validation step and rank by the loss drop. Through experiments on instruction tuning and named entity recognition (NER), they show that ToV typically beats random/uncertainty and often outperforms a strong baseline (LESS), despite requiring less machinery.

**Strengths:**

1 - The core idea is elegantly simple. By exploiting a symmetry between training and validation loss changes (Eq. 6), the method derives a tractable influence proxy that requires only two loss evaluations and a brief “train-on-val” phase, thereby avoiding N validation passes or per-example gradient computations.

2 - They propose two algorithms (Method A/B) and theoretically connect their scores to linearized influence in a gradient descent setting, framing ToV as an influence-style selector that avoids gradients and HVPs.

3 - Empirically, ToV consistently outperforms random and max-uncertainty baselines and, in most cases, exceeds LESS on instruction tuning.

**Weaknesses:**

1 - Because efficiency is central to the paper’s contribution, the claim of practicality should be backed by quantitative evidence, wall-clock time, memory usage, and forward/backward pass counts on identical hardware. LESS has warm-up and gradient-store overhead, whereas ToV involves repeated validation fine-tuning and two full loss sweeps per cycle. A detailed cost comparison would make the “fast” claim more credible.

2 - Since LESS explicitly warns that loss and accuracy may not align in LLM evaluation, relying solely on test log-loss in ToV is limiting. Including complementary metrics, such as accuracy, MMLU-style subsets, or task-specific measures like exact-match or F1, would strengthen the evidence that lower log-loss corresponds to better task quality.

3 - For completeness, ToV should be compared with at least one alignment-based selector (e.g., importance resampling) and a datamodel-style [1] to evaluate whether its advantage persists when competing methods also avoid per-example gradients.

[1] Park et al, "Trak: Attributing model behavior at scale" ICML 2023

**Questions:**

1 - Repeated training on the validation set during scoring raises concerns about overfitting the selection policy. Please clarify how ToV prevents overfitting to $Z^{val}$, and whether performance on the test set deteriorates as the number of ToV cycles increases.

2 - It would be helpful to discuss regimes where the train-on-val = train-on-x symmetry assumption does not hold. Reporting these failure modes or limitations would add practical value and clarify the boundaries of ToV’s effectiveness.

3 - The shift from token-level to sequence-level selection may introduce biases stemming from uneven sequence lengths, as discussed in LESS. Please clarify how ToV accounts for or corrects this bias.

4 - What is the effect of base subset size $m$?

---

> ### Author Response · Authors · 2025-11-21
>
> Thank you for your thoughtful review and for highlighting several key strengths of our paper, including the simplicity and elegance of the proposed approach, and the strong empirical performance of ToV across instruction tuning and NER tasks. We appreciate your constructive feedback and address each point in detail below.
>
> 1. We found ToV to be substantially faster and more memory-efficient across both the instruction-tuning and NER setups. Using the official LESS implementation released by the authors, we measured runtime and memory usage under identical hardware (mean over 5 runs). The results are summarized below:
> | Setting             | Method | Runtime        | Memory Usage                                      |
> |---------------------|--------|----------------|---------------------------------------------------|
> | Instruction Tuning  | LESS   | ~4 hours       | ~4.9 GB (0.9 GB checkpoints + ~4 GB gradients)    |
> | Instruction Tuning  | ToV    | ~2 hours       | ~1.84 GB total (0.141 GB scores)                  |
> | NER                 | LESS   | > 45 minutes   | ~4 GB (gradients dominate)                        |
> | NER                 | ToV    | ~8 minutes     | < 0.25 GB total                                   |
>
> All measurements correspond to Setup 3 for both IT and NER; the results for the remaining setups show the same trend. We will include the full table in the final version of the paper.
>  We also note that both ToV and LESS are far cheaper than directly estimating per-example influences (as discussed in the introduction), while ToV achieves consistently stronger empirical performance.
>
> 2.  While LESS notes that log-loss may not always correlate perfectly with downstream metrics, its own selection objective is also to minimize cross-entropy loss, which is the same objective used by ToV.  In addition, the LESS paper observes that reductions in cross-entropy typically coincide with improvements in accuracy (see point 3 in their limitations section). Since both our method and prior data-selection approaches are explicitly designed to minimize validation cross-entropy, log-loss is the correct metric for evaluating the quality of the selected data. Downstream performance can depend on many additional factors unrelated to the data-selection procedure itself.
> Nonetheless, we agree that reporting complementary metrics can be helpful. For the NER task, we computed tokenwise F1 scores and observed improvements consistent with those in log-loss; we will include these results in the final version. For instruction tuning, there is no single uniform accuracy-style metric across our heterogeneous train and test datasets, so we used log-loss/perplexity, which is the standard evaluation metric for language models. We leave evaluating the improvement in various downstream tasks for future work.
>
> 3. LESS already evaluated several similarity-based alignment baselines and found that they performed poorly, often worse than random, which led us to focus on comparing against the strongest available baseline, namely LESS. A comparison with DataModel-style approaches such as [1] would indeed be valuable, but the authors have not released code for dataset-level selection and the method is known to be sensitive to several hyperparameters, making a reliable comparison difficult. In addition, these methods are much more computationally expensive than both LESS and our approach.
>
> [1] Optimizing ML Training with Metagradient Descent
>
> Answers:
>  1. In Method A, overfitting to the validation set is avoided because each scoring model is trained on the validation data only once. Although we compute scores at multiple checkpoints, each checkpoint uses a model that has not previously been updated on the validation set, so repeated validation updates never accumulate on the same model. This prevents the selection policy from overfitting to $Z^{val}$. Method B may in principle be more susceptible to this issue since it continues training from the validation-updated model, but this can be mitigated by using a validation learning rate that is much smaller than the learning rate used for training on the base set, which is what we do in our experiments.
>
> 2. The symmetry follows from a first-order Taylor approximation, so it holds when the validation update is small and the loss landscape is locally smooth. If the update is too large or the landscape highly non-linear, the approximation may degrade. We choose a small validation learning rate in the experiments to help us keep in the reliable regime.
>
> 3. We also observed the length-related bias in our experiments, and we addressed it by applying binning based on sequence length (see line 211) and then selecting from each bin individually.
>
> 4. Our initial experiments indicate that larger base subset sizes generally improve the performance of the selection methods, as they provide a more stable starting point for scoring. We will include results for different values of $m$ in the final version.

---

> > ### Comment · Reviewer_E8Vn · 2025-11-25
> >
> > Thank you for the detailed response. You addressed all of my concerns except one. It is regarding the loss–accuracy misalignment I noted in Weakness #2.
> >
> > From the supplementary F1 plots you provided during the rebuttal, we can clearly observe a substantial discrepancy between Figure 3 of the paper (log-loss) and the F1 trends. For example, in Figure 3 (exp. 6), all proposed methods outperform random selection by a large margin (20–35%). However, the corresponding F1 plots show the opposite pattern; random selection appears to be the strongest.
> >
> > Given this inconsistency, it becomes difficult to rely on log-loss to make predictions about LESS performance ([as you attempted in your response to Reviewer YYKn](https://openreview.net/forum?id=fWHd3yYicX&noteId=JdeHqowSEs)). Could you clarify how you explain this mismatch between the two metrics?
> >
> > Finally, due to the severity of the discrepancy between log-loss and F1, I believe it is essential to include F1 results for LESS as well, in order to ensure a fair and complete comparison.

---

> > > ### Author Response · Authors · 2025-11-26
> > >
> > > The plot we provided is for 1-F1 score instead of F1 score, so a lower number is better. And the random selection is actually doing the worst, hence there is no discrepancy. We apologize for the confusion.

---

> > > > ### Comment · Reviewer_E8Vn · 2025-11-26
> > > >
> > > > I apologize for the oversight. All of my concerns have now been fully addressed.

---

### Official Review · Reviewer_JbT7 · 2025-11-01

**Soundness:** 3
**Presentation:** 2
**Contribution:** 3
**Rating:** 6
**Confidence:** 2

**Summary:**

The paper tackles data selection from a large training pool, with small datasets for validation and test. The proposed method efficiently scores each example from the training pool by how much one GD step on it contributes to lower validation loss, only with gradients on samples from the validation dataset, not from the training pool, by a clever trick based on first-order Taylor approximation. In section 2, experiments with tasks like instruction tuning and NER show that the proposed method (Method A) is more effective than baselines with respect to log-likelihood loss. Theoretical analysis is provided for a variant of the proposed method (Method B) in Section 3, justifying the proposed method from an asymptotic viewpoint.

**Strengths:**

- The problem addressed in this paper, data selection from a large training pool with a small validation dataset, is important for real-world fine-tuning.
- The derivation of the proposed method, which eliminates the need for computing gradients on the training pool, is straightforward and effective.
- Experimental results indeed show the effectiveness of the proposed method (Method A) compared to baselines in log-likelihood loss.
- Theoretical results provide further justification for their approach beyond just Taylor approximation, based on the asymptotic analysis.

**Weaknesses:**

- The main trick in this paper, i.e., swapping gradients between training and validation data in computing the difference in validation loss after one or multi-steps gradient descent, is not novel itself. Such a trick have already appeared in [Liu et al. 2019] and [Savani et al. 2025] for example. Nevertheless, as far as I know, its application to data selection may be novel.
- I have some concerns on the experimental design in Section 2. While the experiments report only log-likelihood loss, especially in instruction tuning, it does not imply actual improvements for downstream tasks and the instruction-following capability. Also, the choice of the NER task is not reasonable since entity classification for each token is less useful in real world and there are more suitable tasks than NER.
- There is a non-negligible mismatch between the experimental results (focusing on Method A) and the theoretical results (focusing on Method B). Even if Method A is empirically better than Method B, I think Method B should be mainly evaluated also in experiments (or Method A should be mainly analyzed theoretically) for consistency throughout the paper.
- The paper title `Train on Validation` is somewhat misleading. The paper just proposes data selection using gradients on validation data, and actual training is mainly performed on the selected data from the training pool.

[Liu et al. 2019] DARTS: Differentiable Architecture Search (ICLR'19)

[Savani et al. 2025] Antidistillation Sampling, https://arxiv.org/abs/2504.13146

**Questions:**

See weaknesses.

---

> ### Author Response · Authors · 2025-11-21
>
> Thank you for your detailed review and for recognizing the importance of the data-selection problem, the clarity of our derivation, and the value of the theoretical analysis. We address each of your concerns below.
> 1. We believe that DARTS (Liu et al., 2019) is technically very different from our work and Antidistillation Sampling (Savani et al., 2025) is concurrent to our work, and applies related ideas in a very different context.
> DARTS computes hypergradients by differentiating the validation loss through a one-step update on the training loss, which requires gradients on both datasets and a Hessian–vector product. It does not introduce an identity that swaps the roles of training and validation data, nor does it provide a mechanism for scoring individual training examples without per-example gradients. Savani et al. (2025) also use the symmetry of directional derivatives to rewrite an inner product between two gradients, but the setting is entirely different. Their identity is used to adjust a teacher model’s token-level decoding distribution, not to evaluate the influence of training examples. In contrast, our contribution is to apply a train–validation symmetry to re-express the (expensive) influence of a training example on validation loss in terms of the (cheap) change in that example’s loss after a small train-on-validation step.
> Savani et al. (2025) is concurrent to our work, and we thank the reviewer for bringing it to our attention; we will include it in the paper as relevant concurrent work.
>
> 2. Both our data selection method and previous ones are explicitly designed to minimize cross-entropy on the validation set. For this reason, log-loss is the appropriate metric for evaluating the quality of the different methods. While eventually we might be interested in downstream performance, this can be affected by other factors that are not directly related to the quality of data selection by itself. Prior work has also noted that reductions in cross-entropy generally correlate with improvements in downstream quality, and methods such as LESS explicitly use cross-entropy as their selection objective.
> For the NER task, we also computed tokenwise F1 and observed consistent improvements consistent with those in log-loss; we will include these results in the final version. For instruction tuning, there is no single uniform accuracy-style metric across our heterogeneous train/test datasets, so we reported log-loss/perplexity, which is standard for language modeling. We leave evaluating the improvement in various downstream tasks for future work.
> Regarding task choice, our goal was to demonstrate that ToV applies to diverse fine-tuning scenarios, including both generation-style instruction-tuning and token-level structured prediction.
>
> 3. We emphasize that results for Method B are included in the appendix. In our experiments, a simplified version of Method B (Method A) consistently worked better in practice.
> The two methods are closely related and in fact coincide when the number of epochs is one. Method A also avoids repeated updates on the validation set, which helps reduce overfitting in practice.
>
> 4. We appreciate this feedback. Our intention with the title was to highlight the key scoring mechanism that involves briefly training on the validation set.

---

### Meta-Review · Area_Chair_99pH · 2025-12-31

**Summary:**

This paper proposes "Train on Validation" (ToV), a novel and computationally efficient data selection method for fine-tuning large language models. The core idea is to invert the traditional roles of training and validation sets: by observing how predictions on the training pool change after a brief gradient update on a small validation set, the method identifies the most beneficial training examples.

The paper was generally well-received by all four reviewers (all initially scoring a 6: Marginally Above Acceptance). The reviewers praised the method's simplicity, the avoidance of expensive per-example gradients or Hessian-vector products (HVPs), and the strong empirical performance against baselines like LESS. The primary concerns raised during the review process included a lack of quantitative evidence regarding the "fast" efficiency claims, potential misalignment between log-loss and downstream accuracy, and the relationship between the two proposed algorithmic variants (Method A and Method B). The authors provided a comprehensive rebuttal, including specific wall-clock time and memory usage benchmarks, which significantly strengthened the submission.

**Reviewer Concerns:**

#### **Concerns Addressed by the Rebuttal:**
*   **Efficiency Quantification (Reviewers E8Vn, 5rLJ, YYKn):** All reviewers questioned the "fast" claim without specific metrics. The authors provided a detailed table showing that ToV is ~2x faster than LESS for instruction tuning and ~5x faster for NER, while using significantly less memory (e.g., <0.25 GB for NER vs. 4 GB for LESS).
*   **Loss-Accuracy Alignment (Reviewers JbT7, E8Vn, YYKn):** Reviewers were concerned that reporting only log-loss/perplexity was insufficient. The authors clarified that for NER, they observed consistent improvements in F1 scores. They also cleared up a misunderstanding with Reviewer E8Vn regarding a plot that used "1-F1 score" (where lower is better), resolving the perceived discrepancy.
*   **Algorithmic Clarity and Naming (Reviewer YYKn, JbT7):** Reviewers noted a gap between the theoretical analysis (Method B) and the empirical implementation (Method A). The authors addressed this by providing clearer distinctions and renaming the methods to **Interleaved ToV** (Method A) and **Parallel ToV** (Method B) for the final version.
*   **Novelty vs. DARTS/Antidistillation (Reviewer JbT7):** The authors successfully argued that while the "gradient swap" concept appears in other contexts, their application to data selection is distinct and avoids the costly second-order derivatives required by methods like DARTS.
*   **Overfitting to Validation Set (Reviewer E8Vn):** The authors clarified that in "Interleaved ToV," each model branch is updated on validation data only once, preventing the accumulation of validation updates and mitigating overfitting risks.

**Reviewer Scores:**

Reviewer  **E8Vn** may raise the score from 6 to 8 by explicitly stating: "All of my concerns have now been fully addressed" after the F1/Log-loss confusion was resolved.
Other reviewers are more likely to keep their initial score.

---

### Decision · Program_Chairs · 2026-01-26

Accept (Poster)